# TripleSumm: Adaptive Triple-Modality Fusion for Video Summarization

**Sumin Kim**[*]  **Hyemin Jeong**[*]  **Mingu Kang**[*]  **Yejin Kim**  **Yoori Oh**[†]  **Joonseok Lee**[†]
Seoul National University
{sumink, hyeminjeong, gms5560, a2000yejin, yoori0203, joonseok}@snu.ac.kr

## Abstract

The exponential growth of video content necessitates effective video summarization to efficiently extract key information from long videos. However, current approaches struggle to fully comprehend complex videos, primarily because they employ static or modality-agnostic fusion strategies. These methods fail to account for the dynamic, frame-dependent variations in modality saliency inherent in video data. To overcome these limitations, we propose **TripleSumm**, a novel architecture that adaptively weights and fuses the contributions of visual, text, and audio modalities at the frame level. Furthermore, a significant bottleneck for research into multimodal video summarization has been the lack of comprehensive benchmarks. Addressing this bottleneck, we introduce **MoSu** (Most Replayed Multimodal Video Summarization), the first large-scale benchmark that provides all three modalities. Extensive experiments demonstrate that TripleSumm achieves state-of-the-art performance, outperforming existing methods by a significant margin on four benchmarks, including MoSu. Our code and dataset are available at https://github.com/smkim37/TripleSumm.

## 1 Introduction

With recent advances in smart mobile devices and data communication, video content has explosively grown across various platforms such as YouTube or TikTok. At the same time, recent trend shows shifted preference towards short-form content, leading to an increased demand for excerpting the main plot of long videos through summaries. *Video summarization*, the task of extracting key segments that fully represent the content of the original video, serves to meet this demand.

Existing work on video summarization (Apostolidis et al., 2021b; Son et al., 2024; Kim et al., 2025a) has primarily focused on seeking a model architecture mapping frame-level visual features to their importance scores as a summary. However, human comprehension of video is inherently a multimodal process that integrates diverse cues beyond the visual. Most existing architectures, which focus solely on the visual modality, therefore overlook complementary information present in other modalities. Fig. 1, for example, illustrates a music audition that the primary modality to understand the content varies across the video. At the point (a), the text modality (speech) is the most informative to grasp the judge's evaluation, while at the point (b), visual-audio cues play a more important role to enjoy the robot's performance. At the point (c), all three modalities contribute in conjunction. Observing that the modality-specific importance significantly varies even within the same video, we are motivated to utilize multiple modalities in an adaptive manner to dynamically weight the most informative modality for a more effective video summarization.

Recognizing this, recent work has begun to utilize multimodal signals for video summarization (He et al., 2023; Li et al., 2023; Hua et al., 2025; Guo et al., 2025). However, it has been largely underexplored what is the best-suitable way to fuse multimodal signals in the context of video summarization. The core objective of summarization is not merely to predict a true label, but to contrast more important frames to less important ones. This necessitates a discriminative approach to appropriately weigh feature salience by considering two key factors: intra-modal temporal dependency,

---

[*]Equal contribution
[†]Corresponding authors

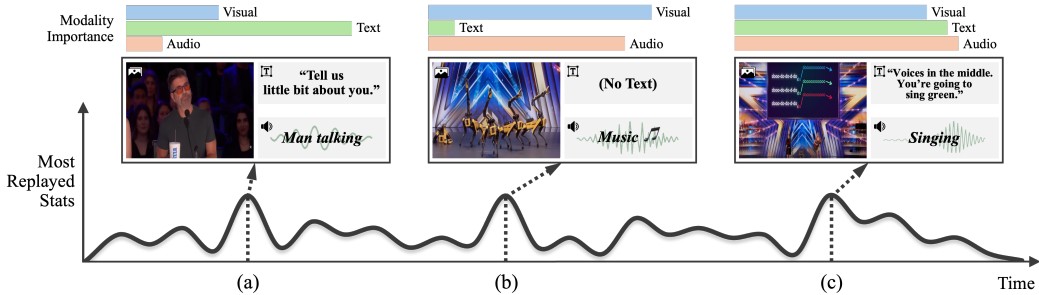

Figure 1: **Illustration of Dynamic Importance of Modalities.** Text is most salient at (a), while visual-audio are dominant at (b). At (c), all three contribute significantly. This highlights the necessity for an adaptive model that dynamically weighs saliency of each modality frame-by-frame.

which involves comparing the same modality features across adjacent time steps, and inter-modal coherence, which involves comparing different modalities at the same time point. Despite using rich trimodal inputs, most current architectures still employ simple or static fusion mechanisms (*e.g.*, standard self-attention or cross-attention) or prioritize a single modality, failing to dynamically focus on the most informative cue at each frame. This results in compromised performance when non-visual cues become highly descriptive.

In this paper, we propose **TripleSumm**, a novel video summarization model that flexibly and adaptively leverages multimodal features, while being robust in the presence of missing modalities, covering all three key aspects of a video: visual, text, and audio. Specifically, TripleSumm employs the following two novel components. First, the *Multi-scale Temporal block* employs a hierarchical sliding window structure with varying window sizes to effectively detect subtle temporal changes without losing the overall narrative of the video. Second, the *Cross-modal Fusion block* incorporates fusion tokens to explicitly learn which modality to prioritize at each moment.

In addition to the model architecture, training data has been another bottleneck for multimodal video summarization. In spite of its importance, there is no publicly available video data equipped with visual-text-audio features and importance score annotations at a sufficient scale. Existing datasets are often in a limited scale (Gygli et al., 2014; Song et al., 2015) or with a limited set of modalities (Narasimhan et al., 2022; Sul et al., 2023; Argaw et al., 2024). To address this, we introduce a large-scale video summarization dataset providing features for all three key modalities, namely, **Mo**st Replayed Multimodal Video **Su**mmarization (**MoSu**). Composed of 52,678 in-the-wild videos and watch behavior aggregated from at least 50,000 viewers per video, this new dataset serves as a highly reliable training and evaluation set for trimodal video summarization.

Through comprehensive experiments, we verify that TripleSumm outperforms existing models by a large margin on multiple benchmarks. Notably, our method robustly generates a reasonable video summary even when one or more modalities are absent, dynamically relying on each modality depending on the content. Our qualitative analysis clearly demonstrates that the proposed method adaptively fuses multimodal information on a frame-by-frame basis.

Our main contributions can be summarized as follows:

- We propose the **TripleSumm** architecture that adaptively fuses visual, text, and audio modalities at frame level. With its temporal and modality blocks, it dynamically adjusts the importance of each modality and effectively captures the micro- and macro-level information of the video.
- We present the **MoSu**, the first large-scale video summarization dataset that provides trimodal features of each video, establishing a reliable foundation for multimodal video summarization.
- We demonstrate that TripleSumm achieves the state-of-the-art on four major video summarization benchmarks, including MoSu, while maintaining parameter efficiency.

## 2  RELATED WORK

**Video Summarization.** Early work in video summarization primarily focused on modeling temporal dependencies within the visual features, utilizing Recurrent Neural Networks or Long Short-term Memory (Zhang et al., 2016; Zhao et al., 2017; 2018; 2021b). More recent models employ a Self-Attention mechanism to capture global, long-range dependencies (Fajtl et al., 2018; Jung et al.,

2019; Zhu et al., 2021; Wang et al., 2020; Apostolidis et al., 2021b; Jiang & Mu, 2022; Terbouche et al., 2023; Son et al., 2024). Additionally, some works leverage graph-based models (Park et al., 2020; Zhao et al., 2021a; Zhu et al., 2022; Zhang et al., 2024), Generative Adversarial Networks (Mahasseni et al., 2017; Yuan et al., 2019; Apostolidis et al., 2019; 2021a), or diffusion models (Yu et al., 2024; Shang et al., 2025; Kim et al., 2025a).

Despite these architectural advances, relying only on visual features limits comprehensive understanding of a video. Shifting towards a multimodal approach, textual data such as transcripts or image captions have been incorporated (Narasimhan et al., 2021; Huang et al., 2021; Narasimhan et al., 2022; Ghauri et al., 2021; Li et al., 2023; Argaw et al., 2024; Qiu et al., 2024), and recently CFSum (Guo et al., 2025) have pioneered to leverage visual, text, and audio signals for summarization. However, the majority of these methods still employ simple or static mechanisms (*e.g.*, standard self-attention or fixed cross-attention modules) to fuse multimodal features, without being fully adaptive. Consequently, they either treat modalities uniformly or prioritize visual information, using text as merely supplementary features (Narasimhan et al., 2021; Li et al., 2023). Due to this inherent bias and lack of dynamic weighting, these models fail to adequately summarize videos where non-visual cues (text or audio) dominate the content. Large Language Models are also employed to capture the multimodal context, leveraging their powerful reasoning capabilities (Lin et al., 2024; Hua et al., 2025; Lee et al., 2025; Kang et al., 2025). Meanwhile, audio remains as a largely under-utilized modality in video summarization, with a few exceptions in recent work (Badamdorj et al., 2021; Liu et al., 2022; Zhao et al., 2022; Xie et al., 2022). Our work addresses the need for a balanced utilization of all three primary modalities (visual, text, and audio) achieving comprehensive video summarization.

**Video Summarization Datasets.** SumMe (Gygli et al., 2014) and TVSum (Song et al., 2015) have been widely used, but they suffer from two critical limitations: extremely small scale (25 and 50 videos, respectively) and unimodal annotation based solely on visual cues, making them suboptimal for multimodal research. A large-scale benchmark Mr. HiSum (Sul et al., 2023) addresses the scale issue, but its deliberate exclusion of audio-rich categories such as music leaves the modality gap unaddressed. More recent datasets have focused on incorporating textual information. However, they often introduce new challenges, such as domain bias towards instructional videos (*e.g.*, (Argaw et al., 2024)) or live streams (*e.g.*, (He et al., 2023)), or suffer from sparse ground truths (*e.g.*, (Qiu et al., 2024)). We address the need for a large-scale, trimodal and diverse dataset by introducing the MoSu dataset, detailed in Sec. 4.

## 3 TripleSumm: The Proposed Method

We present our video summarization model, **TripleSumm**, depicted in Fig. 2. Beginning with the multimodal feature extraction (Sec. 3.1), we subsequently describe how we fuse the temporal and cross-modal information using our two core components, *Multi-scale Temporal block* and *Cross-modal Fusion block* (Sec. 3.2). Lastly, we describe the inference process to predict the frame-level importance scores and to generate the final summary video from them (Sec. 3.3).

### 3.1 Input Representation

A raw video consists of multiple modality streams with different sampling rates. We preprocess these streams to produce a set of synchronized feature sequences, each resampled to $N$ time steps. (See Sec. 5.1 for details.) At each time step $i = 1, ..., N$, we are given multimodal signals. Under our trimodal setting, we assume that we have visual $\mathcal{V} = \{\mathbf{V}_1, ..., \mathbf{V}_N\}$, text $\mathcal{T} = \{\mathbf{T}_1, ..., \mathbf{T}_N\}$, and audio $\mathcal{A} = \{\mathbf{A}_1, ..., \mathbf{A}_N\}$, where $\mathbf{V}_i$, $\mathbf{T}_i$, and $\mathbf{A}_i$ are the modality-specific raw data; *e.g.*, $\mathbf{V}_i$ is a 2D RGB image. We will describe our method based on this trimodal setting, but our method itself can be extended to an arbitrary set of multimodal features.

For each item in each modality sequence, we extract feature representations, denoted by $\mathbf{X}^{\{v,t,a\}}$, employing modality-specific pre-trained encoders:

$$\mathbf{X}^v = \phi(\mathbf{V}) \in \mathbb{R}^{N \times D_v}, \ \ \mathbf{X}^t = \psi(\mathbf{T}) \in \mathbb{R}^{N \times D_t}, \ \ \mathbf{X}^a = \omega(\mathbf{A}) \in \mathbb{R}^{N \times D_a}, \tag{1}$$

where $\phi$ is an image encoder (*e.g.*, (Szegedy et al., 2015; Radford et al., 2021)), $\psi$ is a text encoder (*e.g.*, (Liu et al., 2019)), and $\omega$ is an audio encoder (*e.g.*, (Gong et al., 2021)). Each pre-trained

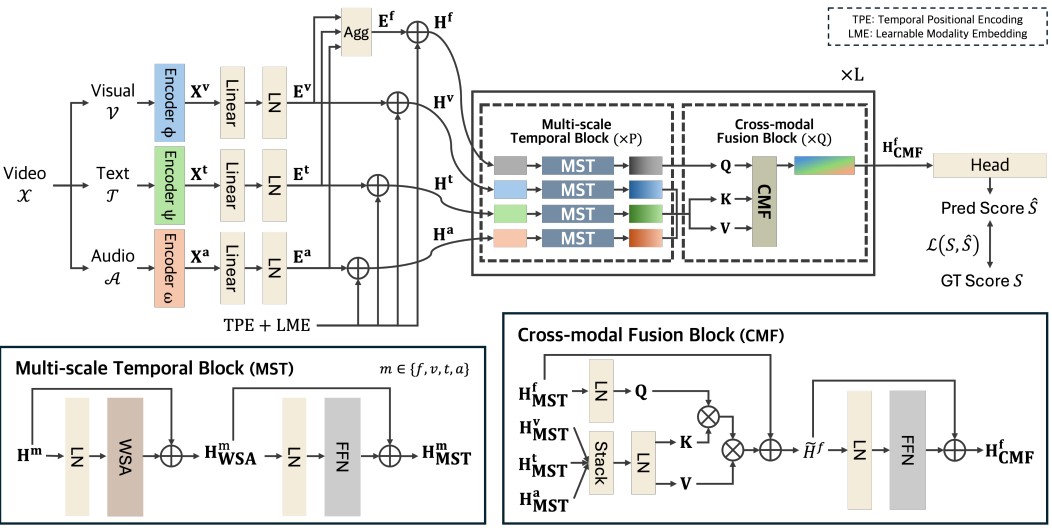

Figure 2: **Overall architecture of TripleSumm.** Visual, text, and audio features are first encoded and linearly projected, then aggregated into fusion tokens, refined through the **Multi-scale Temporal block** (MST, lower left), and fused in the **Cross-modal Fusion block** (CMF, lower right). The fused representation is passed through a prediction head to generate frame-level importance scores.

encoder produces an embedding of its own size, denoted by $D_v$, $D_t$, and $D_a$ for visual, text, and audio, respectively.

These modality-specific features reside in different latent spaces. To effectively fuse them in subsequent modules, we project them into a common embedding space of size $D$. For this, we apply a linear projection (Linear) and layer normalization (LN) (Ba et al., 2016) to each modality-specific features $\mathbf{X}^{\{v,t,a\}}$ to produce per-modality embeddings $\mathbf{E}^{\{v,t,a\}} \in \mathbb{R}^{N \times D}$:

$$\mathbf{E}^{\{v,t,a\}} = \text{LN}(\text{Linear}(\mathbf{X}^{\{v,t,a\}})). \tag{2}$$

We denote the embedding at time step $i$, which is the $i$-th row of $\mathbf{E}^{\{v,t,a\}}$, as $\mathbf{e}_i^{\{v,t,a\}}$. On top of these per-modality embeddings, we introduce another cross-modal embedding, denoted by $\mathbf{E}^f = \{\mathbf{e}_1^f, \ldots, \mathbf{e}_N^f\} \in \mathbb{R}^{N \times D}$, where $\mathbf{e}_i^f = \text{Agg}(\mathbf{e}_i^v, \mathbf{e}_i^t, \mathbf{e}_i^a)$. The aggregation function (Agg) can be either a deterministic function (*e.g.*, average), or a learnable model (*e.g.*, multi-layer perceptrons). The main motivation to add this is to avoid potential bias introduced by conventional cross-modal fusion methods, *e.g.*, asymmetric using one modality as a query to attend on others. Our fused embeddings $\mathbf{E}^f$ serve as an anchor to integrate all modalities, promoting their equitable engagement.

Lastly, we construct the final token embeddings by adding a temporal positional encoding (TPE) $\mathbf{tpe}_i \in \mathbb{R}^D$ (Vaswani et al., 2017) and a learnable modality embedding (LME) $\mathbf{lme}^{\{f,v,t,a\}} \in \mathbb{R}^D$ to distinguish each time step and the origin of the modality:

$$\mathbf{h}_i^{\{f,v,t,a\}} = \mathbf{e}_i^{\{f,v,t,a\}} + \mathbf{tpe}_i + \mathbf{lme}^{\{f,v,t,a\}}, \quad \text{for } i = 1, \ldots, N. \tag{3}$$

In matrix form, the final input matrix stacked over all time steps is denoted by $\mathbf{H}^{\{f,v,t,a\}} \in \mathbb{R}^{N \times D}$.

## 3.2 TEMPORAL AND CROSS-MODAL REFINEMENT

The core of our proposed architecture is a hierarchical 'refine-and-fuse' strategy, designed to learn integrated representations from the input sequences ($\mathbf{H}^{\{f,v,t,a\}}$). This is achieved by interleaving two key components: Multi-scale Temporal block (MST) for temporal refinement within each modality, and Cross-modal Fusion block (CMF) for inter-modal information exchange. We stack $L$ interleaved layers, where each layer is composed of $P$ temporal blocks and $Q$ cross-modal blocks.

**Multi-scale Temporal Block.** From the input sequence $\mathbf{H}^{\{f,v,t,a\}}$, the MST learns temporal patterns within each modality by employing Windowed Self-Attention (WSA) (Beltagy et al., 2020; Liu et al., 2021), restricting the range of self-attention for a query token $\mathbf{q}_i \in \mathbb{R}^D$ at time step $i$ to keys and values within a local window of size $w$ centered at $i$. These local keys and values, denoted

by $\mathbf{K}_i \in \mathbb{R}^{w \times d_k}$ and $\mathbf{V}_i \in \mathbb{R}^{w \times d_k}$, are constructed by collecting key vectors $\mathbf{k}_j$ and value vectors $\mathbf{v}_j$ within a window such that $j = |i - j| \leq w$, where $w \in \{n \in \mathbb{N} \mid 1 \leq n \leq N$ and $n$ is odd$\}$ and $d_k$ is the dimensionality of the key vector. Formally, the overall process of the MST is as follows:

$$\mathbf{h}_{\text{WSA}}^m = \text{Attn}(\text{LN}(\mathbf{h}_i^m), \text{LN}(\mathbf{h}_{\{j:|i-j| \leq w\}}^m), \text{LN}(\mathbf{h}_{\{j:|i-j| \leq w\}}^m)) + \mathbf{h}^m,$$

$$\mathbf{h}_{\text{MST}}^m = \text{FFN}(\text{LN}(\mathbf{h}_{\text{WSA}}^m)) + \mathbf{h}_{\text{WSA}}^m, \tag{4}$$

where $m \in \{f, v, t, a\}$, $\text{Attn}(\mathbf{Q}, \mathbf{K}, \mathbf{V}) = \text{softmax}\left(\mathbf{Q}\mathbf{K}^\top / \sqrt{d_k}\right)\mathbf{V}$ is the standard attention (Vaswani et al., 2017), and FFN is a feed-forward network. The 'Multi-scale' characteristic is achieved by varying the window size $w$ at each layer, to allow the model to capture semantics from local to global scales. Specifically, the initial layers employ smaller window sizes to capture fine-grained local dependencies between adjacent frames. Then, later layers progressively expand the windows to capture long-range dependencies across frames. WSA is applied to each modality with shared parameters, enabling the model to capture general temporal patterns while keeping parameter usage efficient. An ablation on parameter sharing is detailed in the App. C.5. Furthermore, WSA is not just crucial for the model to capture multi-granular context across the input video, it is also beneficial for computational efficiency, reducing the complexity from $O(N^2)$ for standard self-attention to $O(w \cdot N)$ at each layer.

**Cross-modal Fusion Block.** Whereas the MST focuses on refining temporal patterns within each modality, the CMF is designed to model the interactions across different modalities, independently at each time step. To allow the model to select the most informative modality at each time step without being biased towards any particular modality, this block employs the cross-attention mechanism taking the fusion token $\mathbf{h}_i^f$ as a single query at each time step $i$, and the corresponding modality-specific tokens, $\mathbf{h}_i^{\{v,t,a\}}$, as the keys and values. The query token then attends to this collection of three context tokens, allowing it to weigh and aggregate information from the most relevant modality at that specific moment.

Formally, given the features $\mathbf{h}_{\text{MST},i}^{\{f,v,t,a\}}$ from the preceding temporal block at time step $i$, it performs

$$\tilde{\mathbf{h}}_i^f = \text{Attn}(\text{LN}(\mathbf{h}_{\text{MST},i}^f), \text{LN}(\mathbf{h}_{\text{MST},i}^{\{v,t,a\}}), \text{LN}(\mathbf{h}_{\text{MST},i}^{\{v,t,a\}})) + \mathbf{h}_{\text{MST},i}^f, \tag{5}$$

$$\mathbf{h}_{\text{CMF},i}^f = \text{FFN}(\text{LN}(\tilde{\mathbf{h}}_i^f)) + \tilde{\mathbf{h}}_i^f, \tag{6}$$

where $\text{Attn}(\mathbf{Q}, \mathbf{K}, \mathbf{V}) = \text{softmax}\left(\mathbf{Q}\mathbf{K}^\top / \sqrt{d_k}\right)\mathbf{V}$, and $d_k$ is the key vector size. The Layer Normalization $\text{LN}(\cdot)$ is applied within each modality separately. The updated fused embeddings, $\mathbf{H}_{\text{CMF}}^f$, are enriched with contextual information selectively drawn from all three modalities.

A key design principle of TripleSumm is the separation of temporal and cross-modal fusion. The temporal context is modeled exclusively by the MST, allowing it to focus solely on integrating modality information at each corresponding time step. On the other hand, the cross-modal fusion is solely performed by the CMF, at each time step $i$ independently. This design is not just beneficial for the model to explicitly learn the two orthogonal (temporal and multimodal) patterns in the video, but also allows efficient implementation by momentarily merging the batch and time dimensions, enabling parallel processing across all tokens.

## 3.3 MODEL TRAINING AND INFERENCE

Finally, a prediction head linearly projects the refined fused features $\mathbf{H}_{\text{CMF}}^f$ to the importance score $\hat{S} \in [0, 1]$ of each frame, interpreted as the probability to be included in the final video summary. More details about the prediction head are described in App. A.1.

The model is trained to minimize the squared L2 loss between the predicted score vector $\hat{S} = \{\hat{s}_1, \ldots, \hat{s}_N\}$ and the ground-truth scores $S = \{s_1, \ldots, s_N\}$. Formally, our loss is given by

$$\mathcal{L}(S, \hat{S}) = \|S - \hat{S}\|_2^2 = \|S - \text{Linear}(\mathbf{H}_{\text{CMF}}^f)\|_2^2. \tag{7}$$

Following established practice in video summarization (Otani et al., 2019; Son et al., 2024; Kim et al., 2025a), the final summary is obtained by selecting a set of temporally coherent shots that maximizes the predicted frame-level importance scores under a fixed length budget. The detailed segmentation and selection procedure is provided in App. A.2.

## 4 MoSu Dataset

Video summarization has been suffering from the data shortage for both quality and scale. For example, the most widely used public datasets for the last decade were SumMe (Gygli et al., 2014) and TVSum (Song et al., 2015), which consist of just 25 and 50 videos, respectively. Obviously, these are in an extremely small scale, led serious overfitting environment, but no other dataset has been available until recently due to the high labeling cost. Mr. HiSum (Sul et al., 2023) resolved this by taking the 'Most Replayed' statistics from YouTube as a reliable proxy for per-frame importance based on collective viewer engagement (see App. B.2 for details). However, despite this advantage, Mr. HiSum remains unsuitable for trimodal video summarization, as it provides only visual features without accompanying text or audio modalities and additionally removes audio-centric video categories, further limiting its applicability to multimodal fusion.

In order to reliably train and evaluate trimodal video summarization models, we introduce a new large-scale called dataset **MoSu** (**Mo**st Replayed Multimodal Video **Su**mmarization), curated from YouTube-8M (Abu-El-Haija et al., 2016), which provides videos with multi-label annotations across 3,862 classes drawn from knowledge graph entities. For annotations, we follow a similar procedure to Mr. HiSum (Sul et al., 2023), collecting the 'Most Replayed' statistic. We construct this dataset by filtering videos that satisfy the following criteria: 1) both an English transcription and an audio track are available, either originally provided or automatically generated by YouTube's caption translation, to satisfy the trimodal condition, 2) over 50,000 views to obtain the Most Replayed statistics, and 3) at least 120 seconds long to ensure sufficiently long content for meaningful summarization.

**Dataset Statistics.** MoSu contains 52,678 videos corresponding to nearly 4,000 hours, covering a vast range of topics with 3,406 categories. The average video length in MoSu is 272.3 seconds, ranging from 120 to 501 seconds. As compared with other video summarization datasets in Tab. 1, MoSu is the first large-scale dataset to provide all three modalities (visual, text, and audio), as opposed to previous ones with visual-only (*e.g.*, Mr. HiSum) or bimodal (*e.g.*, MMSum).

| Dataset | Modality | | | Videos | Duration | Category |
|---------|--------|------|-------|--------|----------|----------|
| | Visual | Text | Audio | | | |
| SumMe | ✓ | | | 25 | 1.1 | – |
| TVSum | ✓ | | | 50 | 3.5 | 10 |
| Mr.HiSum | ✓ | | | 31,892 | 1,788.0 | 3,509 |
| TL;DW? | ✓ | ✓ | | 12,160 | 628.5 | 185 |
| BLiSS | ✓ | ✓ | | 13,303 | 1,109.0 | – |
| LfVS-P | ✓ | ✓ | | 250K | 55,416.6 | 6,700 |
| MMSum | ✓ | ✓ | | 5,100 | 1,229.9 | 170 |
| **MoSu** | ✓ | ✓ | ✓ | 52,678 | 3,983.7 | 3,406 |

Table 1: **Statistics of various video summarization datasets.** Total duration is reported in hours.

**Thematic Categories.** Motivated by the finding that the complexity of video summarization significantly varies by video topics (Sul et al., 2023), we aim to analyze video summarization quality across various thematic categories on MoSu as well. Since the original 3,406 categories inherited from YouTube-8M are too fine-grained, we cluster them into 10 distinct topical groups. Specifically, we first construct a topic space by applying K-Means clustering to the SBERT embeddings (Reimers & Gurevych, 2019) of the 3,862 entity descriptions from Wikipedia. We also map each video to the same embedding space using its concatenated entity labels, and select the closest cluster centroid as its pseudo-class. See App. B.3 for details of the resulting thematic groups; App. G.2 for summarization performance across clusters, showing how they affect the ease of summarization.

## 5 Experiments

### 5.1 Experimental Setting

**Datasets.** Our primary evaluation is conducted on our MoSu dataset, introduced in Sec. 4. To further validate generalizability of the proposed model, we also evaluate its performance on three widely-used external datasets: a large-scale Mr. HiSum (Sul et al., 2023), and two human-annotated SumMe (Gygli et al., 2014) and TVSum (Song et al., 2015). We follow the original data split for Mr. HiSum and MoSu. For SumMe and TVSum, we evaluate under a 5-fold cross-validation framework using two distinct protocols: the traditional train/validation (TV) split without a test set (Li et al., 2023; Son et al., 2024; Lee et al., 2025), and a train/validation/test (TVT) split (Kim et al., 2025a) to correct the overfitting issues inherent in the TV setup. Details for protocols are available in App. B.5.

| Method | Modality V | T | A | $\tau\uparrow$ | $\rho\uparrow$ | mAP50↑ | mAP15↑ | Params↓ | GFLOPs↓ |
|---|---|---|---|---|---|---|---|---|---|
| VASNet (Fajtl et al., 2018) | ✓ | | | 0.151 | 0.219 | 64.49 | 31.05 | 8.13M | 1.99G |
| PGL-SUM (Apostolidis et al., 2021b) | ✓ | | | 0.151 | 0.218 | 64.97 | 30.63 | 5.31M | 1.21G |
| CSTA (Son et al., 2024) | ✓ | | | 0.291 | 0.398 | 71.77 | 40.65 | 10.56M | 11.37G |
| A2Summ (He et al., 2023) | ✓ | ✓ | | 0.181 | 0.257 | 66.48 | 35.70 | 2.48M | 1.35G |
| SSPVS (Li et al., 2023) | ✓ | ✓ | | 0.190 | 0.271 | 66.10 | 32.65 | 112.81M | 43.64G |
| Joint-VA (Badamdorj et al., 2021) | ✓ | | ✓ | 0.190 | 0.272 | 65.68 | 32.25 | 4.21M | 1.63G |
| UMT (Liu et al., 2022) | ✓ | | ✓ | 0.239 | 0.334 | 68.83 | 36.73 | 4.66M | 1.39G |
| CFSum (Guo et al., 2025) | ✓ | ✓ | ✓ | 0.277 | 0.374 | 70.97 | 38.20 | 19.83M | 8.52G |
| **TripleSumm (Ours)** | ✓ | ✓ | ✓ | **0.351** | **0.472** | **74.72** | **44.42** | **1.37M** | **0.97G** |

Table 2: **Comparison on MoSu.** Best and second-best are **boldfaced** and underlined, respectively.

| (a) Mr. HiSum | | | | | (b) SumMe | | | | | | (c) TVSum | | | | |
|---|---|---|---|---|---|---|---|---|---|---|---|---|---|---|---|
| | TVT | | | | | TVT | | TV | | | TVT | | TV | | |
| **Method** | $\tau$ | $\rho$ | mAP50 | mAP15 | **Method** | $\tau$ | $\rho$ | $\tau$ | $\rho$ | | $\tau$ | $\rho$ | $\tau$ | $\rho$ | |
| VASNet | 0.069 | 0.102 | 58.69 | 25.28 | VASNet | 0.089 | 0.099 | 0.160 | 0.170 | | 0.153 | 0.205 | 0.160 | 0.170 | |
| PGL-SUM | 0.097 | 0.141 | 61.60 | 27.45 | PGL-SUM | 0.104 | 0.116 | 0.192 | 0.213 | | 0.141 | 0.186 | 0.157 | 0.206 | |
| CSTA | 0.128 | 0.185 | 63.38 | 30.42 | CSTA | 0.133 | 0.148 | 0.246 | 0.274 | | 0.168 | 0.221 | 0.194 | 0.255 | |
| A2Summ | 0.121 | 0.172 | 63.20 | 32.34 | A2Summ | 0.088 | 0.096 | 0.108 | 0.129 | | 0.157 | 0.206 | 0.137 | 0.165 | |
| SSPVS | 0.078 | 0.113 | 59.48 | 26.35 | SSPVS | 0.142 | 0.157 | 0.192 | 0.257 | | 0.171 | 0.226 | 0.181 | 0.238 | |
| Joint-VA | 0.161 | 0.231 | 65.88 | 35.23 | Joint-VA | 0.117 | 0.129 | 0.230 | 0.256 | | 0.142 | 0.188 | 0.166 | 0.220 | |
| UMT | 0.178 | 0.253 | 66.81 | 35.65 | UMT | 0.148 | 0.165 | 0.241 | 0.268 | | 0.144 | 0.189 | 0.179 | 0.235 | |
| **Ours** (Visual) | 0.187 | 0.258 | 67.16 | 35.57 | **Ours** (Full) | 0.162 | 0.187 | 0.265 | 0.296 | | 0.198 | 0.259 | 0.211 | 0.275 | |
| **Ours** (Full) | **0.258** | **0.352** | **70.72** | **40.88** | **Ours** (MoSu) | **0.172** | **0.192** | **0.282** | **0.314** | | **0.200** | **0.262** | **0.217** | **0.282** | |

Table 3: **Performance comparison on (a) Mr. HiSum, (b) SumMe, and (c) TVSum.** Ours(Full) refers to the model trained on each respective dataset. Ours(Visual) is trained using only visual features, and Ours(MoSu) is pre-trained on MoSu and fine-tuned on each target dataset.

**Data Preprocessing.** As mentioned in Sec. 3.1, our model requires temporally aligned feature sequences of the same length. We preprocess each modality in MoSu as follows. For the visual modality, we sample frames at 1 fps and encode them using a pre-trained CLIP (Radford et al., 2021). For the text modality, we extract time-stamped transcripts from YouTube and obtain RoBERTa (Liu et al., 2019) features by taking the sentence-level [CLS] token, which is then broadcast to all frames covered by its duration; frames without transcripts are filled with a default embedding vector. For audio, we extract features at a 1-second interval using a centered-window approach: for each second $t$, a 10-second segment is cropped from the interval $[t-5, t+5]$ and encoded into a feature vector using a pre-trained Audio Spectrogram Transformer (Gong et al., 2021).

On Mr. HiSum, SumMe, and TVSum, we adopt their officially provided visual features for a fair comparison. Since these datasets do not provide text or audio streams, we generate frame-level text captions using an image captioning model, Qwen2.5-VL-7B-Instruct (Bai et al., 2025), and extract the raw audio directly from the videos. For each modality, we then follow the preprocessing procedures originally defined for each dataset. Detailed preprocessing steps are provided in App. B.4.

**Implementation Details.** We use $L = 2$ interleaved layers, with $P = 2$ Multi-scale Temporal blocks (thus, in total 4 temporal blocks) and $Q = 2$ Cross-modal Fusion block. To facilitate a local-to-global feature capture, we progressively increase the attention window size ($w$), from 5 up to the entire sequence length $N$. More details are described in App. A.1.

**Evaluation Metrics.** Following Otani et al. (2019), we primarily report rank-based correlation metrics, Kendall's $\tau$ (Kendall, 1945) and Spearman's $\rho$ (Zwillinger & Kokoska, 1999), on the frame-level importance prediction, being consistent with recent studies (Son et al., 2024; Terbouche et al., 2023; Kim et al., 2025a). On MoSu and Mr. HiSum datasets where the Most Replayed statistics are used as ground-truth, we further assess the highlight detection performance with mean Average Precision (mAP) following Sul et al. (2023). Following this protocol, each video is divided into 5-second segments and ranked by predicted importance. The top 50% (mAP50) and 15% (mAP15) of these segments are then evaluated as the predicted highlights against the ground-truth.

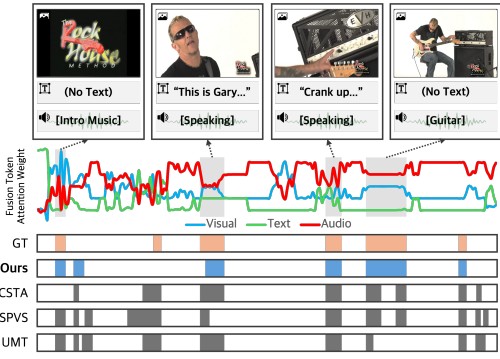 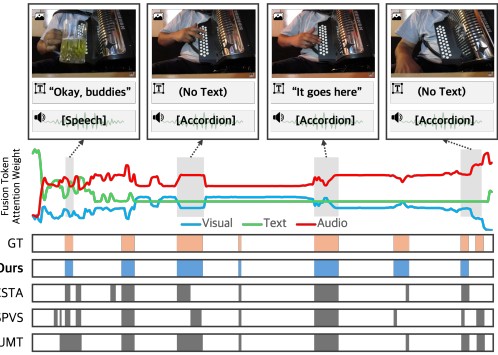

(a) Guitar demonstration where attention shifts among visual, text, and audio cues as the content changes.

(b) Playing accordion video with less visual-text information, where audio cues predominate.

Figure 3: **Qualitative Examples on MoSu.** The graph in the middle visualizes the fusion token's attention weights, illustrating how our model dynamically estimates the saliency of each modality and thus maintains strong summarization accuracy even when some modalities are missing.

## 5.2 QUANTITATIVE EVALUATION

**Evaluation on MoSu.** As shown in Tab. 2, TripleSumm achieves clear state-of-the-art performance on MoSu, surpassing all unimodal and multimodal baselines by a substantial margin across all metrics. In addition to its accuracy gains, TripleSumm is also highly efficient with only 1.37M parameters, which is significantly smaller than strong baselines such as CSTA (10.56M), and UMT (4.66M). The results demonstrate that TripleSumm delivers superior predictive performance while maintaining a markedly lighter and more efficient architecture.

**Evaluation on Other Datasets.** Tab. 3(a) shows that our method achieves state-of-the-art performance on most metrics on Mr. HiSum. Notably, it outperforms all baselines even only with the visual features Ours$_{(Visual)}$, validating the effectiveness of the proposed Multi-scale Temporal block in capturing salient temporal features from a single modality. In addition, our full model significantly outperforms its visual-only version, highlighting the benefit of leveraging multimodal information.

Tab. 3(b–c) indicate that our full model outperforms all baselines by a large margin when trained and evaluated on SumMe and TVSum, both of which are human-annotated datasets. This is particularly noteworthy, since the ground-truth annotations are solely based on visual information. Our model's performance with all three modalities suggests that non-visual modalities provide crucial contextual cues for identifying key visual moments, implying that our trimodal approach learns a synergistic relationship among modalities. Lastly, our model performs even better when pre-trained on MoSu and fine-tuned on the target dataset (the last row, Ours$_{(MoSu)}$). This result indicates that the rich representations learned from the large-scale MoSu dataset are effectively transferable.

## 5.3 QUALITATIVE DEMONSTRATION

We illustrate a couple of qualitative examples of the video summary with TripleSumm in Fig. 3. As seen in the graph in the middle, showing the attention weights of the *Fusion Token*, TripleSumm adapts the saliency of each modality on a frame-by-frame basis as the content changes. Through these weights, we interpret that our method has properly learned and reflected the relative importance of each modality, aligned well with human intuition, instead of exhibiting chaotic fluctuations or collapsing to a static mean. In a guitar demonstration video in Fig. 3a, the model relies mostly on audio signals, while correctly shifting attention to the visual modality for the opening logo (first scene) and to text for the narration (second scene). Fig. 3b shows an accordion performance with minimal visual changes and no text. Despite missing modalities, it maintains strong predictions by properly attending to the audio. This capability stems from our design using the neutral Fusion Token, not a modality-specific one, as the query, allowing the model to weigh modalities free from bias. Overall, we observe that TripleSumm produces summaries more faithful to the ground truth than the baselines by appropriately weighing modalities at each moment. More qualitative results are available in App. H.

**(a) Input Modalities**

| V | T | A | $\tau$ | $\rho$ | mAP50 | mAP15 |
|---|---|---|---|---|---|---|
| ✓ | | | 0.298 | 0.404 | 72.20 | 40.24 |
| | ✓ | | 0.267 | 0.364 | 70.01 | 37.37 |
| | | ✓ | 0.271 | 0.370 | 69.96 | 37.88 |
| | ✓ | ✓ | 0.307 | 0.416 | 72.41 | 41.20 |
| ✓ | ✓ | | 0.328 | 0.442 | 73.77 | 42.39 |
| ✓ | | ✓ | _0.331_ | _0.448_ | _73.85_ | _42.64_ |
| ✓ | ✓ | ✓ | **0.351** | **0.472** | **74.72** | **44.42** |

**(b) Attention Window Size**

| Window Size ($w$) | $\tau$ | $\rho$ | mAP50 | mAP15 |
|---|---|---|---|---|
| Constant (Local) | 0.297 | 0.409 | 71.61 | 39.73 |
| Constant (Wide) | 0.304 | 0.416 | 72.54 | 40.26 |
| Constant (Global) | 0.317 | 0.431 | 73.46 | 41.62 |
| Wide-to-Local | 0.325 | 0.441 | 73.20 | 42.13 |
| Local-to-Wide | 0.335 | 0.455 | 73.83 | 43.41 |
| Global-to-Local | _0.345_ | _0.461_ | _74.36_ | _43.61_ |
| **Local-to-Global** | **0.351** | **0.472** | **74.72** | **44.42** |

**(c) MST & CMF Block**

| MST | CMF | $\tau$ | $\rho$ | mAP50 | mAP15 |
|---|---|---|---|---|---|
| | ✓ | 0.250 | 0.352 | 70.01 | 37.35 |
| ✓ | | _0.337_ | _0.452_ | _73.90_ | _43.08_ |
| ✓ | ✓ | **0.351** | **0.472** | **74.72** | **44.42** |

**(d) Modality Fusion**

| Method | $\tau$ | $\rho$ | mAP50 | mAP15 |
|---|---|---|---|---|
| Static | 0.338 | 0.458 | 74.08 | 43.31 |
| Global | _0.344_ | _0.461_ | _74.26_ | _43.59_ |
| **Dynamic** | **0.351** | **0.472** | **74.72** | **44.42** |

Table 4: **Ablation studies on input modalities, attention window sizes, MST & CMF blocks, and modality fusion methods.**

## 5.4 ABLATION STUDY

We further conduct several ablation studies on MoSu, unless noted otherwise.

**Ablation on Input Modalities.** We evaluate our method with all possible combinations of input modalities in Tab. 4(a). Among the unimodal settings, the visual modality (V) turns out to be strongest, while the audio (A) slightly outperforms the text (T). This highlights the power of the audio stream, which continuously provides rich contextual cues, whereas transcripts are often sparse in "in-the-wild" videos, leaving many segments without semantic information. Each bimodal combination outperforms the strongest single modality, demonstrating a significant synergistic effect. Building on this synergy, the full trimodal configuration achieves the best performance, confirming that all three modalities provide unique and valuable signals for summarization.

**Ablation on Window Size.** We compare several hierarchical windowing options for the multi-scale temporal blocks. We denote $w \in \{3, 5, 7\}$ as Local windows and its 3-9 times larger size as Wide windows. A Global window indicates $w = N$, where $N$ is the total number of frames in the video. Using these window sizes, we experiment with various scheduling strategies: using a constant window size, and gradually expanding or shrinking. For instance, Local-to-Global means starting with a Local window size at the first layer, and adopts gradually larger sizes in the next layers, reaching to the whole frames at the last one; *e.g.*, [3, 9, 27, $N$] or [5, 15, 45, $N$].

We first observe in Tab. 4(b) that relying on a constant window size yields suboptimal performance. Here, the Constant (Global) is equivalent to the standard self-attention, using all frames at all layers, but it underperforms the following mixed strategies. Among the hierarchical ones, we observe two patterns. First, narrower-to-wider strategies (Local-to-Wide/Global) tend to outperform its opposite (Wide/Global-to-Local), demonstrating that a bottom-up approach to learn fine-grained details first and establish broader context based on them is more appropriate for video summarization. Also, having a Globally-windowed layer slightly improves performance for both bottom-up and top-down strategies, at the cost of computational overhead. We draw these conclusions based on more fine-grained experiments with various choices of $w$, detailed in App. C.2.

**Ablation on MST & CMF Blocks.** We conduct an ablation study to verify the effect of the two components of our method: the Multi-scale Temporal (MST) block and the Cross-modal Fusion (CMF) block. The results in Tab. 4(c) clearly show that both blocks are essential to achieve the full performance of our model, observing a significant performance drop when one of them is removed. Comparing the effect of the two components, we observe a more substantial degradation when MST is removed. This highlights that the MST block is the most critical module for accurately capturing the long-range temporal dependencies and multi-scale context required for precise frame-level importance prediction. In conclusion, the CMF block indeed provides the final multimodal boost by dynamically fusing the processed features, and the MST block forms the foundational backbone essential for leveraging the complex, long-range temporal structure of the video, confirming their respective roles in our proposed architecture.

**Ablation on Modality Fusion Method.** A main hypothesis of our paper is that the saliency of each modality varies frame by frame. To verify its validity, we conduct an ablation study on its design. The simplest approach to fuse trimodal tokens would be simply taking average over them with same weights (*Static*). A slightly advanced variant would learn a single scalar weight for each modality by averaging cross-modal attention scores over time (*Global*). These global weights are then uniformly applied across the video, allowing adaptation across modalities but not across frames. Finally, *Dynamic* fusion computes cross-attention scores independently at every frame and uses these scores to determine the relative contribution of modality features on a per-frame basis.

As shown in Tab. 4(d), the performance gradually improves as modality and temporal adaptivity are added (*Dynamic > Global > Static*). This result confirms that the model effectively leverages frame-level flexibility without collapsing to a trivial solution, validating that our *Dynamic* fusion mechanism is stable and essential for effective multimodal summarization.

## 5.5 Zero-shot Performance on Long-form Videos

For a more rigorous evaluation of the scalability and generalization capabilities of our model, we conduct another zero-shot experiment on 50 significantly longer videos unseen at training. On average, these test videos are 70.4 minutes long, covering a wide range of topics. More details about this test set are provided in App. B.6.

All models have been trained on MoSu, and Tab. 5 reports their zero-shot inference performance on this long video set. The results clearly demonstrate the superior generalization of TripleSumm, particularly in the demanding long-form setting. On these extremely long videos, our model significantly outperforms all baselines on the rank-based correlation metrics. Specifically, our TripleSumm

| Method | $\tau$ | $\rho$ | mAP50 | mAP15 |
|---|---|---|---|---|
| Random | 0.000 | 0.000 | 50.59 | 16.18 |
| VASNet | 0.024 | 0.036 | 54.15 | 18.95 |
| PGL-SUM | 0.024 | 0.035 | 54.29 | 17.55 |
| CSTA | 0.083 | 0.123 | 58.09 | 22.26 |
| A2Summ | 0.042 | 0.062 | 54.04 | 18.66 |
| SSPVS | 0.033 | 0.048 | 53.60 | 18.45 |
| Joint-VA | 0.052 | 0.077 | 53.99 | 19.90 |
| UMT | 0.066 | 0.097 | 56.05 | 23.10 |
| CFSum | 0.061 | 0.089 | 56.32 | 20.59 |
| **TripleSumm** | **0.128** | **0.189** | **59.70** | **23.27** |

Table 5: **Zero-shot performance on long videos.** All models are trained on MoSu and tested directly on long video dataset.

achieves the highest scores in Kendall's $\tau$ (0.128) and Spearman's $\rho$ (0.189). This confirms that our adaptive fusion architecture generalizes far more effectively to complex, long-form content, maintaining the most accurate frame-level importance prediction even when faced with entirely new domains and semantic structures.

To the best of our knowledge, this is the first summarization benchmark on hours-long videos. Considering the fact that video summarization gets more challenging but meaningful on these longer videos with more complex story-telling structures, we believe that this experiment also significantly contributes the to community in advancing video summarization in real-world scenarios.

## 6 Conclusion

We primarily explore a deep integration of three modalities, visual, text, and audio, for the task of video summarization. Our proposed method, **TripleSumm**, dynamically assesses the saliency of each modality at different moments and adaptively utilizes them to produce a superior summary video. To facilitate further development, we also introduce **MoSu**, a new large-scale trimodal dataset. Our experiments show that TripleSumm achieves highly competitive results on multiple benchmarks. These findings suggest the potential benefits of integrating all three modalities for a more comprehensive understanding of video content. We hope that the proposed TripleSumm model and the MoSu dataset can contribute to future advancements in multimodal video summarization.

While our work adheres to the standard three-step protocol (frame importance scoring, temporal segmentation, and segment selection) for fair comparison on existing benchmarks, we believe that developing a fully end-to-end trainable model would be a promising future direction. Exploring methods that learn to select coherent summary clips directly, rather than only learning frame-level scores, would present a valuable opportunity to advance the field.

## ACKNOWLEDGMENTS

This work was also supported by the SOFT Foundry Institute at SNU, Samsung Electronics, Youlchon Foundation, National Research Foundation of Korea (NRF) grants (RS-2021-NR05515, RS-2024-00336576, RS-2023-0022663, RS-2025-25399604, RS-2025-25421642), and the Institute for Information & Communication Technology Planning & Evaluation (IITP) grants (RS-2022-II220264, RS-2024-00353131) funded by the Korean government.

## LARGE LANGUAGE MODEL USAGE

We acknowledge the use of large language models to enhance readability, correct grammar, and refine sentence structure during the preparation of this paper. All intellectual contributions, including the core ideas, methodology, and analysis, are solely the work of the authors.

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

APPENDIX

## A  IMPLEMENTATION DETAILS

### A.1  EXPERIMENTAL SETUP

The model's hyperparameters are detailed in Tab. I. The architecture consists of 2 interleaved layers with 2 multi-scale temporal blocks and 2 cross-modal fusion blocks. Totally, the architecture consists of 4 multi-scale temporal blocks. To capture features from local-to-global scales, the attention window size $w$ is progressively increased across these blocks, from local self-attention (a window size of 5) up to 15, 45, and $N$, where $N$ is the input sequence length. We use the Swish-Gated Linear Unit (SwiGLU) (Shazeer, 2020) for the Feed-Forward Network (FFN) layers. The final score prediction head comprises a sequence of a linear projection, a GeLU activation, layer normalization, another linear projection, and a final sigmoid activation. The hidden dimension within this head is set to 192. Architectural parameters include an embedding dimension ($D$) of 128. The model was trained for 100 epochs using the AdamW optimizer. The learning rate was initialized at $1 \times 10^{-4}$ and adjusted via a Cosine scheduler. All experiments were conducted on a single NVIDIA RTX A100.

| Category | Hyperparameter | Value |
|---|---|---|
| *Model Architecture* | Embedding Dimension ($D$) | 128 |
| | Number of multi-scale temporal block ($P$) | 2 |
| | Number of cross-modal fusion block ($Q$) | 2 |
| | Number of interleaved block ($L$) | 2 |
| | Number of attention heads | 4 |
| | Hidden dimension of prediction head | 192 |
| *Training Details* | Epoch | 100 |
| | Batch Size | 64 |
| | Dropout rate | 0.1 |
| | Initial Learning Rate | $1 \times 10^{-4}$ |

Table I: **Hyperparameters for the proposed model.**

### A.2  SUMMARY GENERATION PROCEDURE

To construct the final summary from the predicted per-frame importance scores $\hat{S}$, we follow the standard pipeline used in prior work (Son et al., 2024; Kim et al., 2025a). Each input video is first partitioned into temporally coherent shots using Kernel Temporal Segmentation (KTS) (Potapov et al., 2014; Zhang et al., 2016). The importance score of each shot is then computed as the mean of the predicted frame-level scores within that shot. Next, we select an optimal subset of shots that maximizes the total shot score while satisfying a predefined length budget (*e.g.*, 15% of the original video duration); this selection is formulated as a 0/1 knapsack problem with shot length as the weight and shot score as the value. Finally, the chosen shots are concatenated in their original temporal order to produce the final summary video.

## B  DATASETS DETAILS

### B.1  DETAILED STATISTICS OF THE MOSU DATASET

We provide more detailed characteristics of the MoSu dataset, introduced in Sec. 4. First, Fig. I shows the distribution of video duration in this dataset, ranging from 120 to 500 seconds. Tab. II offers statistics of the video duration, text tokens, and audio tracks. As we filter out all videos without a valid audio track (*e.g.* a scenario that can occur if a creator mutes or removes the audio), all videos in this dataset have audio information, making it suitable for triple modality fusion.

| Statistic Category | Value |
|---|---|
| *Duration Statistics* | |
| Avg. Duration | 272.25 sec |
| Std. Deviation | 102.43 sec |
| Min Duration | 120.00 sec |
| Max Duration | 501.00 sec |
| *Textual Statistics* | |
| Total # of Tokens | 32.6M |
| Avg. # of Tokens per Video | 619.1 |
| Transcript Density | 61.84% |
| *Audio Statistics* | |
| Audio Availability | 100% |

Table II: **Detailed Statistics of the MoSu Dataset.** Transcript density means the average ratio of video duration with valid text.

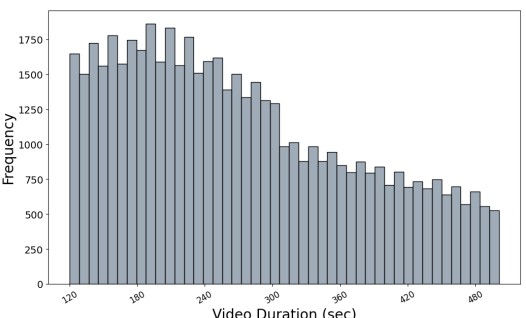

Figure I: **Video Durations in MoSu Dataset.**

## B.2 GROUND-TRUTH PREPROCESSING

The 'Most Replayed' statistic from YouTube serves as the ground-truth for identifying key moments in a video. It is a normalized frequency of replay counts collected from tens of thousands of views, provided as a sequence of 100 scores corresponding to uniformly segmented clips. Following the precedent set by Sul et al. (2023); Kim et al. (2025b), we exclusively use videos with over 50,000 views to ensure statistical reliability.

**Handling YouTube Most Replayed Bias.** A common artifact was observed in these statistics: an anomalous spike in replay scores within the initial few seconds of a video, often occurring before the primary content begins. This phenomenon, referred to as YouTube Most Replayed Bias, is likely attributable to viewers' heightened attention or initial buffering adjustments at the start of a video. If left unaddressed, this bias could mislead a model into associating high importance with temporal position rather than semantic content. To mitigate this effect and ensure the model learns content-based importance, the ground-truth scores corresponding to the first five seconds of each video were zeroed out.

## B.3 THEMATIC CATEGORY CLUSTERING

Tab. III shows the number of videos and the top five most frequent entity labels for each identified cluster. A high degree of semantic coherence is observed between the top entities and assigned cluster topics, which validates the effectiveness of the clustering approach. The dataset was partitioned into train, validation, and test sets by applying a stratified split with an 8:1:1 ratio to each cluster. This ensures that the proportional representation of each cluster is preserved across all data subsets.

## B.4 DATA PREPROCESSING DETAILS

To ensure reproducibility, we provide the detailed settings for our feature extraction pipeline, which processes all videos into temporally aligned feature sequences at a 1-second interval.

**Visual Features.** For our MoSu dataset, we sampled one frame per second and used the official pre-trained CLIP model[1] to extract a 768-dimensional feature vector for each frame. For the external benchmarks, to ensure a fair comparison, we directly utilized their officially provided visual features: Inception-v3 features for Mr. HiSum, and GoogLeNet features for SumMe and TVSum.

**Text Features.** The text features were derived from two sources: YouTube transcripts for the MoSu dataset and generated captions for external benchmarks. For the MoSu dataset, time-stamped transcripts provided by YouTube were utilized. English transcripts were used directly, while non-English ones were translated via YouTube's automated translation API. Videos without any transcripts were excluded. For external benchmarks, a caption was generated for each frame using a Vision-Language

---

[1] openai/clip-vit-large-patch14

| Cluster | Videos | Top 5 Most Frequent Entities |
|---|---|---|
| Video Games | 11,578 | Game, Video game, Action-adventure game, Call of Duty, Strategy video game |
| Musical Instruments | 7,647 | Guitar, String instrument, Musician, Acoustic guitar, Electric guitar |
| Fashion & Materials | 5,588 | Cosmetics, Hair, Hairstyle, Fashion, Art |
| Automobiles | 5,207 | Vehicle, Car, Engine, Driving, Sports car |
| Electronic Devices | 4,291 | Gadget, Mobile phone, Smartphone, Samsung Galaxy, IPhone |
| Food & Cooking | 3,919 | Food, Recipe, Cooking, Dessert, Dish (Food) |
| Sports | 3,784 | Game, Weight training, Basketball, Gym, Basketball moves |
| Animation & Comics | 3,752 | Cartoon, Comedy (drama), Animation, Art, Comedian |
| Vehicles & Transportation | 3,731 | Vehicle, Motorcycle, Cycling, Bicycle, Motorcycling |
| Animals | 3,181 | Animal, Pet, Dog, Fishing, Fish |

Table III: Cluster sizes and their top 5 most frequent entities.

model[2]. All acquired text was then subjected to a unified encoding process. First, non-verbal annotations (*e.g.*, [Music], [Applause]) were removed. Each sentence was subsequently encoded into a 768-dimensional feature vector using the `[CLS]` token embedding from a pre-trained RoBERTa model[3]. This vector was then broadcasted across all 1-second timestamps spanned by its duration. Timestamps with no associated text were assigned a default vector, precomputed by passing a `<pad>` token through the encoder.

**Audio Features.** To process the audio data, we extracted a feature vector for each second of the audio stream. This is achieved using a centered window approach designed to capture sufficient semantic context and match the model's standard input size. Specifically, for each second $t$, a 10-second segment is extracted from the symmetric interval $[t-5, t+5]$. This bidirectional windowing ensures that the representation for time $t$ is informed by both preceding and succeeding events. These segments are subsequently encoded into a fixed-size feature vector by utilizing the output embedding of the `[CLS]` token from a pre-trained Audio Spectrogram Transformer model[4]. This process results in a sequence of contextualized representations for the entire audio stream.

### B.5 DATA SPLIT IN SUMME AND TVSUM

For completeness and reproducibility, we provide detailed descriptions of the two evaluation protocols used in the main paper. Both protocols operate under a 5-fold cross-validation framework but differ significantly in how the data is partitioned for model development and evaluation.

**Traditional Train/Validation (TV) Split.** Following standard practice in earlier video summarization research, the dataset is randomly partitioned into five equal folds. For each iteration, four folds are used for training and the remaining one fold is used for evaluation, with the final results averaged across all five folds. However, this protocol lacks a separate, dedicated test set. Consequently, hyperparameter tuning and early stopping decisions are effectively made based on the performance of the held-out evaluation fold, making the model highly susceptible to overfitting.

**Train/Validation/Test (TVT) Split.** To address the overfitting concerns inherent in the traditional TV setup, we also evaluate our method using a stricter TVT split within the 5FCV framework, as adopted by Kim et al. (2025a). In this setting, the five folds are explicitly divided into training, validation, and test sets (typically utilizing three folds for training, one for validation, and one for testing, approximating a 6:2:2 ratio). The validation fold is strictly used for hyperparameter tuning and early stopping, ensuring that the test fold remains completely unseen during the model development phase. This protocol provides a more rigorous and reliable evaluation of the model's true generalization capability.

---

[2] Qwen/Qwen2-VL-7B-Instruct

[3] FacebookAI/roberta-base

[4] MIT/ast-finetuned-audioset-10-10-0.4593

## B.6 LONG VIDEO TEST SET

In order to evaluate the video summarization models in a more realistic and challenging setting, we collect a dedicated evaluation set with significantly longer videos than any other public datasets for video summarization. Traditional video summarization benchmarks like SumMe (Gygli et al., 2014) and TVSum (Song et al., 2015) include videos with a duration of 2-5 minutes on average. Recent datasets like LfVS-P (Argaw et al., 2024) and MMSum (Qiu et al., 2024) have encreased it to 13-14 minutes on average, as shown in Tab. 1. However, these datasets are still far from long videos in the real-world, *e.g.*, movies or documentaries that are hours-long.

We collected 50 diverse, long-form videos curated to challenge models with domains not present in the MoSu training set. While MoSu covers a wide range of categories (*e.g.*, Video Games, Fashion, Animals), this dataset features complex narrative structures and specialized content. For example, it includes domains such as full-length films and documentaries, professional technical tutorials, full-length sports matches and multilingual talk show. Ground truth for these videos have been obtained using the same 'Most Replayed' scores, same as MoSu. Key statistics of this new test set are summarized in Tab. IV. Notably, the average video duration is 70.4 minutes, 10-20 times longer than the training videos from MoSu.

| Statistic | Duration (sec) |
|---|---|
| Avgerage Duration | 4224.0 |
| Stdandard Deviation | 1263.6 |
| Min Duration | 2413.0 |
| Max Duration | 7207.0 |

Table IV: **Statistics for the long video dataset.**

## C ARCHITECTURE ABLATION STUDY

### C.1 MODEL LAYER

| Number of Layers | | | $\tau \uparrow$ | $\rho \uparrow$ | mAP50 $\uparrow$ | mAP15 $\uparrow$ | Params $\downarrow$ |
|---|---|---|---|---|---|---|---|
| Overall (L) | MST (P) | CMF (Q) | | | | | |
| 8 | 8 | 8 | 0.326 | 0.442 | 73.41 | 42.2 | 17.15M |
| 4 | 4 | 4 | 0.346 | 0.465 | 74.39 | 43.34 | 4.53M |
| **2** | **2** | **2** | **0.351** | **0.472** | **74.72** | **44.42** | 1.37M |
| 2 | 2 | 1 | 0.330 | 0.451 | 73.67 | 42.96 | 1.11M |
| 2 | 1 | 2 | 0.339 | 0.461 | 74.41 | 43.33 | 1.11M |
| 2 | 1 | 1 | 0.340 | 0.461 | 74.38 | 43.40 | 0.85M |
| 1 | 1 | 1 | 0.314 | 0.430 | 73.4 | 41.61 | **0.58M** |

Table V: **Ablation study on the number of model layers.** Overall denotes the total number of layers, while MST and CMF refer to the Multi-Scale Temporal block and Cross-Modal Fusion block, respectively.

To investigate the influence of network depth, we vary both the total number of layers and the allocation of layers between the Multi-scale Temporal (MST) block and the Cross-modal Fusion (CMF) block, as reported in Tab. V. Increasing the number of CMF layers provides no additional benefit and results in a slight decline in overall performance across all evaluation metrics, whereas decreasing the MST depth consistently degrades performance. These results indicate that temporal modeling requires a moderately deep MST for effective representation, while a single fusion layer is sufficient for cross-modal integration.

### C.2 WINDOW SIZE

To analyze the impact of the temporal receptive field on summarization performance, we experiment with various window size configurations for the window-based self-attention mechanism, as shown

| Window Size ($w$) | $\tau$ ↑ | $\rho$ ↑ | mAP50 ↑ | mAP15 ↑ |
|---|---|---|---|---|
| **Baselines** | | | | |
| Standard SA | 0.317 | 0.431 | 73.46 | 41.62 |
| **Fixed Window** | | | | |
| $5, 5, 5, 5$ | 0.297 | 0.409 | 71.61 | 39.73 |
| $15, 15, 15, 15$ | 0.318 | 0.430 | 72.64 | 41.15 |
| $45, 45, 45, 45$ | 0.322 | 0.435 | 72.95 | 42.10 |
| **Global-to-Local** | | | | |
| $N, 27, 9, 3$ | 0.340 | 0.462 | 74.22 | 43.09 |
| $N, 45, 15, 5$ | 0.345 | 0.461 | 74.36 | 43.61 |
| $N, 63, 21, 7$ | 0.347 | 0.468 | 74.52 | 43.54 |
| **Local-to-Global** | | | | |
| $3, 9, 27, N$ | 0.346 | 0.466 | 74.58 | 43.91 |
| $\mathbf{5, 15, 45, N}$ | **0.351** | **0.472** | **74.72** | **44.42** |
| $7, 21, 63, N$ | 0.348 | 0.468 | 74.67 | 43.92 |

Table VI: **Ablation study on different window size configurations.** $N$ denotes the full sequence length (global attention).

in Tab. VI. Specifically, we compare fixed window sizes and hierarchical configurations (Global-to-Local and Local-to-Global). The results demonstrate that the Local-to-Global strategy, specifically the sequence of $[5, 15, 45, N]$, yields the best performance across all metrics. This configuration progressively expands the receptive field from the lower to higher layers, allowing the model to capture fine-grained local temporal dynamics in the early stages and integrating global context in the later stages. Conversely, fixed small windows fail to capture long-range dependencies, while starting with global attention (Global-to-Local) appears less effective at preserving essential local details. These findings confirm that a hierarchically expanding window structure is optimal for modeling the multi-scale nature of video content.

## C.3 LEARNABLE WINDOW

| Window Size ($w$) | $\tau$ ↑ | $\rho$ ↑ | mAP50 ↑ | mAP15 ↑ |
|---|---|---|---|---|
| Standard SA | 0.317 | 0.431 | 73.46 | 41.62 |
| + *Learnable* | 0.318 (+0.001) | 0.429 (-0.002) | 73.65 (+0.19) | 41.44 (-0.18) |
| Global-to-Local | 0.345 | 0.461 | 74.36 | 43.61 |
| + *Learnable* | 0.344 (-0.001) | 0.465 (+0.004) | 74.49 (+0.13) | 43.77 (+0.16) |
| **Local-to-Global** | 0.351 | **0.472** | 74.72 | **44.42** |
| + *Learnable* | **0.353 (+0.002)** | 0.469 (-0.003) | **74.83 (+0.11)** | 44.32 (-0.10) |

Table VII: **Effect of Learnable Gaussian modulation on different attention strategies.**

Ideally, it would be the best to adaptively choose the set of window sizes from the data. However, learning these from scratch was extremely unstable from our initial experiments, probably due to the excessive degree of freedom. Instead, we adopt a learnable window configuration based on a Gaussian-modulated self-attention mechanism, conceptually similar to (Guo et al., 2019), initializing the model with a pre-trained one with fixed window sizes. The core idea is to set the parameters of the Gaussian distribution (including the standard deviation $\sigma$) as learnable values, allowing the model to adaptively adjust the width (or scale) of the receptive field at each layer. Specifically, by modulating the attention scores with the Gaussian mask $\mathbf{G}_{i,j}$, the parameter $\sigma$ directly controls the decay rate of attention weights with respect to the temporal distance $|i - j|$. Mathematically, a smaller $\sigma$ causes the exponential term to vanish rapidly for distant frames, effectively enforcing locality, whereas a larger $\sigma$ yields a flatter distribution, allowing the model to capture global context.

As shown in Tab. VII, this adaptive mechanism does not yield a substantial difference. Its effect is negligible across all configurations, including Standard SA, Global-to-Local, and our Local-to-Global strategy, with some metrics even degrading. This suggests that our manually designed

Local-to-Global structure is already highly effective at capturing the necessary multi-scale temporal dependencies. The additional complexity of learning the receptive field variance does not offer a tangible benefit over our carefully tuned fixed windows.

## C.4 EMBEDDING DIMENSION

| $D$ | $\tau\uparrow$ | $\rho\uparrow$ | mAP50 $\uparrow$ | mAP15 $\uparrow$ | Params $\downarrow$ | GFLOPs $\downarrow$ |
|---|---|---|---|---|---|---|
| 768 | 0.234 | 0.324 | 68.25 | 34.50 | 39.71M | 31.77G |
| 512 | 0.313 | 0.424 | 72.40 | 40.84 | 18.07M | 14.25G |
| 384 | 0.344 | 0.464 | 74.44 | 43.18 | 10.42M | 8.10G |
| 256 | 0.347 | 0.467 | 74.54 | 44.1 | 4.85M | 3.67G |
| 192 | 0.350 | 0.471 | 74.59 | 43.84 | 2.85M | 2.10G |
| **128** | **0.351** | **0.472** | **74.72** | **44.42** | 1.37M | 0.97G |
| 96 | 0.348 | 0.467 | 74.56 | 43.8 | **0.84M** | **0.57G** |

Table VIII: **Ablation study on different embedding dimensions.** $D$ denotes the embedding dimension.

We further investigate the effect of the embedding dimension $D$ on both model efficiency and summarization accuracy. As presented in Tab. VIII, we observe that the model performs best around $D = 128$ to 256. The performance slightly degrades up to $D = 384$, and an even larger $D$ substantially overfits the model. Even at an extremely low dimension of $D = 96$, the model maintains competitive performance, demonstrating the robustness and compactness of our feature representation. Based on these results, we select $D = 128$ as the optimal setting, striking a good balance between a lightweight architecture and high-quality summarization.

## C.5 SHARED PARAMETERS

| Setting | $\tau\uparrow$ | $\rho\uparrow$ | mAP50 $\uparrow$ | mAP15 $\uparrow$ | Params $\downarrow$ |
|---|---|---|---|---|---|
| w/o Shared Parameters | 0.343 | 0.459 | 74.24 | 43.24 | 2.95M |
| **w/ Shared Parameters** | **0.351** | **0.472** | **74.72** | **44.42** | **1.37M** |

Table IX: **Effect of shared parameters on performance.**

To evaluate the impact of parameter sharing in the Multi-scale Temporal block, we perform an ablation study (Tab. IX). The w/o Shared Parameters configuration assigns an independent temporal block to each modality, whereas the w/ Shared Parameters configuration employs a single block shared across visual, text, and audio streams.

Sharing parameters reduces the number of learnable weights from 2.95 M to 1.37 M, nearly a three-fold decrease, yet yields consistently higher scores on all evaluation metrics. This result indicates that parameter sharing enables the temporal block to capture multi-scale patterns common to different modalities. Training this shared block on all modalities simultaneously also exposes it to roughly four times as many training sequences as an independent block for each modality. The combination of greater data exposure and the inherent ability of the multi-scale temporal structure to model both long- and short-range dynamics provides a clear explanation for the superior performance of the shared-parameter design, despite its substantially smaller parameter count.

## C.6 FUSION TOKENS

We explore three ways to aggregate trimodal features $(\mathbf{e}_i^v, \mathbf{e}_i^t, \mathbf{e}_i^a)$ into the fusion token $\mathbf{e}_i^f$: 1) a visual-centric baseline, taking only the visual feature ($\mathbf{e}_i^f = \mathbf{e}_i^v$; *No Fusion*), 2) treating the fusion token itself as a learnable parameter ($\mathbf{e}_i^f \in \mathbb{R}^D$; *Learnable*) and 3) a simple average over the visual, text, and audio features ($\mathbf{e}_i^f = (\mathbf{e}_i^v + \mathbf{e}_i^t + \mathbf{e}_i^a)/3$; *Average*).

Surprisingly, Tab. 4(d) concludes that the simple *Average* initialization yields the highest performance. The *No Fusion* baseline, relying mainly on visual information, may be vulnerable when

non-visual cues such as speech or ambient sound dominate or when certain modalities are missing. The *Average*, assigning equal contribution to all modalities, is simple but effective to balance all modalities to serve as a query.

| Method | $\tau\uparrow$ | $\rho\uparrow$ | mAP50 $\uparrow$ | mAP15 $\uparrow$ |
|---|---|---|---|---|
| No Fusion | 0.337 | 0.451 | 73.96 | 43.48 |
| Learnable | 0.341 | 0.462 | 74.41 | 43.71 |
| **Average** | **0.351** | **0.472** | **74.72** | **44.42** |

Table X: **Ablation study on fusion token aggregation.**

## D    COMPUTATIONAL COST ANALYSIS

| Method | $\tau\uparrow$ | $\rho\uparrow$ | Params $\downarrow$ | GFLOPs $\downarrow$ | Inference Time $\downarrow$ |
|---|---|---|---|---|---|
| VASNet | 0.151 | 0.219 | 8.31M | 1.99G | 7.36ms |
| PGL-SUM | 0.151 | 0.218 | 5.31M | 1.21G | 13.55ms |
| CSTA | 0.291 | 0.398 | 10.56M | 11.37G | 9.48ms |
| A2Summ | 0.181 | 0.257 | 2.48M | 1.35G | 44.29ms |
| SSPVS | 0.190 | 0.271 | 112.81M | 43.64G | 14.27ms |
| Joint-VA | 0.190 | 0.272 | 4.21M | 1.63G | **2.58ms** |
| UMT | 0.239 | 0.334 | 4.66M | 1.39G | 4.94ms |
| CFSum | 0.277 | 0.374 | 19.83M | 8.52G | 3.87ms |
| **TripleSumm** | **0.351** | **0.472** | **1.37M** | **0.97G** | 2.81ms |

Table XI: **Comparison of computational cost and performance.** We report rank correlation scores ($\tau$, $\rho$), model size (Params), inference cost (GFLOPs) and inference time to evaluate the trade-off between efficiency and accuracy.

We compare TripleSumm with other methods in terms of computational complexity and inference speed to validate its suitability for real-world applications. Tab. XI reports the summarization accuracy ($\tau$ and $\rho$), number of learnable parameters (Params), GFLOPs and inference time of competing models. TripleSumm sets a new state-of-the-art in efficiency, requiring only 1.37M learnable parameters and 0.97 GFLOPs, which is significantly lower than all other baselines. Despite its lightweight, our model outperforms all baselines by a large margin in correlation metrics ($\tau = 0.361$, $\rho = 0.484$). Furthermore, TripleSumm achieves a remarkably fast inference time of 2.81ms, making it comparable to the fastest method (Joint-VA) while delivering superior summarization quality. This analysis confirms that TripleSumm successfully breaks the trade-off between performance and efficiency, offering a highly scalable solution for video summarization.

## E    ADDITIONAL ANALYSIS ON ADAPTIVE LEARNING DYNAMICS

### E.1    ABLATION ON THE LEARNED MODALITY SALIENCY

To verify if our model effectively assigns attention weights to the truly important modalities at each frame, we conduct another ablation study on the MoSu test set. At inference, we utilize the attention weights assigned by the *Fusion Token* to the three modalities (Visual, Text, and Audio) at every frame. We then rank them in the order of weights: highest (Rank 1), second-highest (Rank 2), and third-highest attention weights (Rank 3), respectively, at each time step. Using these ranks, we selectively take only a subset of modalities (*e.g.*, "Rank 1 only" or "Rank 1 + 2").

As shown in Tab. XII, the full TripleSumm model, which always utilizes all modalities, achieves the best performance across all metrics, confirming that the weighted contribution of all three modalities is the optimal strategy. As expected, the performance gradually degrades as the model is forced to ignore more highly-weighted modalities, showing worst performance when restricted to the least-important modality ("Rank 3 only"). This demonstrates that the attention mechanism successfully suppresses noise and prioritizes informative cues. The substantial performance improvement from

"Rank 1 only" to "Rank 1 + 2" validates that the second-most important modality provides crucial complementary information. In conclusion, this ablation study confirms that the Fusion Token's attention mechanism successfully learns the frame-dependent saliency of each modality, leading to the superior performance of TripleSumm.

| Method | $\tau \uparrow$ | $\rho \uparrow$ | mAP50 $\uparrow$ | mAP15 $\uparrow$ |
|---|---|---|---|---|
| Rank 3 only | 0.058 | 0.087 | 59.44 | 25.20 |
| Rank 2 only | 0.136 | 0.194 | 63.34 | 29.99 |
| Rank 2 + 3 | 0.158 | 0.220 | 64.06 | 30.81 |
| Rank 1 only | 0.196 | 0.273 | 66.09 | 33.28 |
| Rank 1 + 2 | 0.273 | 0.373 | 70.28 | 38.69 |
| **TripleSumm (Ours)** | **0.351** | **0.472** | **74.72** | **44.42** |

Table XII: **Ablation on dropping modality features based on estimated saliency.**

### E.2 ANALYSIS OF ATTENTION WEIGHT DISTRIBUTION

To investigate the distribution of the attention weights learned by the *Fusion Token*, we conduct another ablation study similar to the above (App. E.1), but using a specific threshold $\theta$ instead of relative importance. This experiment is designed to assess whether the learned weights are heavily concentrated on a few dominant modalities or if they are distributed more broadly across the input features.

As depicted in Fig. II, the analysis utilized two filtering strategies. "Over $\theta$" retains only the weights $\geq \theta$ and sets others to zero), while "Under $\theta$" retains only the weights $\leq \theta$ and sets others to zero). The performance metrics, Kendall's $\tau$ and Spearman's $\rho$, exhibit a remarkably smooth and linear inverse relationship as the threshold $\theta$ is varied from $0.0$ to $1.0$. This linearity suggests that the attention weights are not concentrated on a small, fixed subset of features. If the weights are sparse, we would expect to see sharp, non-linear changes in performance. Instead, the gradual degradation of "Over $\theta$" performance and the corresponding gradual improvement of "Under $\theta$" performance (as $\theta$ increases and includes more subtle weights) indicate that the Fusion Token distributes its attention weights broadly and subtly. This broad distribution confirms that the model captures complementary information from a wide range of features and modalities, validating that TripleSumm effectively utilizes distributed saliency rather than relying on a single dominant cue.

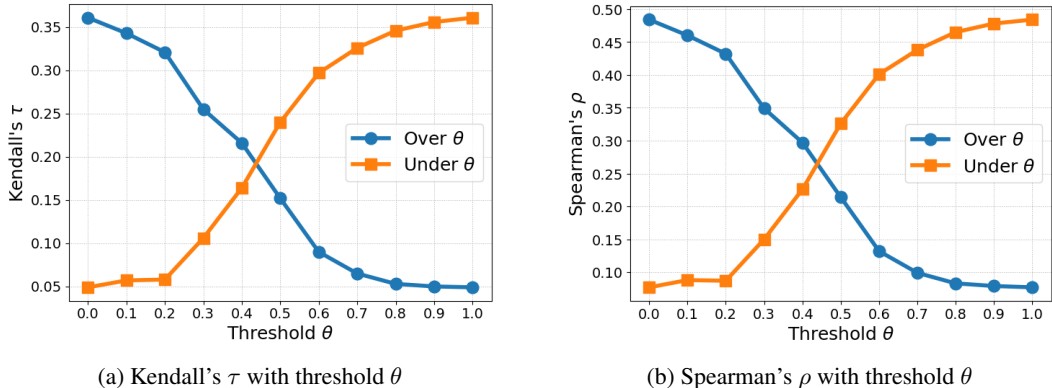

(a) Kendall's $\tau$ with threshold $\theta$    (b) Spearman's $\rho$ with threshold $\theta$

Figure II: **Performance evaluation by thresholding modality attention weights.** The figure illustrates the model performance, measured by (a) Kendall's $\tau$ and (b) Spearman's $\rho$, when a threshold $\theta$ is applied to the Fusion Token's learned modality attention weights. The **Over** $\theta$ line shows the performance when only weights $\geq \theta$ are retained, while the **Under** $\theta$ line shows performance when only weights $\leq \theta$ are retained.

# F   FULL RESULTS OF EXTERNAL DATASET

## F.1   TVSUM AND SUMME

| Method | SumMe | | TVSum | |
|---|---|---|---|---|
| | $\tau \uparrow$ | $\rho \uparrow$ | $\tau \uparrow$ | $\rho \uparrow$ |
| Random | 0.000 | 0.000 | 0.000 | 0.000 |
| Human | 0.205 | 0.213 | 0.177 | 0.204 |
| SUM-GAN(Mahasseni et al., 2017) | 0.049 | 0.066 | 0.024 | 0.031 |
| VASNet(Fajtl et al., 2018) | 0.160 | 0.170 | 0.160 | 0.170 |
| DSNet-AF(Zhu et al., 2021) | 0.037 | 0.046 | 0.113 | 0.138 |
| DSNet-AB(Zhu et al., 2021) | 0.051 | 0.059 | 0.108 | 0.129 |
| AC-SUM-GAN(Apostolidis et al., 2021a) | 0.102 | 0.088 | 0.031 | 0.041 |
| DMASum(Wang et al., 2020) | 0.063 | 0.089 | 0.203 | 0.267 |
| CLIP-It(Narasimhan et al., 2021) | - | - | 0.108 | 0.147 |
| PGL-SUM(Apostolidis et al., 2021b) | 0.192 | 0.213 | 0.157 | 0.206 |
| UMT(Liu et al., 2022) | 0.241 | 0.268 | 0.179 | 0.235 |
| iPTNet(Jiang & Mu, 2022) | 0.101 | 0.119 | 0.134 | 0.163 |
| A2Summ(He et al., 2023) | 0.108 | 0.129 | 0.137 | 0.165 |
| AAAM(Terbouche et al., 2023) | - | - | 0.169 | 0.223 |
| MAAM(Terbouche et al., 2023) | - | - | 0.179 | 0.236 |
| VSS-Net(Zhang et al., 2024) | - | - | 0.190 | 0.249 |
| Joint-VA(Badamdorj et al., 2021) | 0.230 | 0.256 | 0.166 | 0.220 |
| SSPVS(Li et al., 2023) | 0.192 | 0.257 | 0.181 | 0.238 |
| CSTA(Son et al., 2024) | 0.246 | 0.274 | 0.194 | 0.255 |
| LLMVS(Lee et al., 2025) | 0.253 | 0.282 | 0.211 | 0.275 |
| SummDiff(Kim et al., 2025a) | 0.256 | 0.285 | 0.195 | 0.255 |
| **Ours**(Full) | 0.265 | 0.296 | 0.211 | 0.275 |
| **Ours**(MoSu) | **0.282** | **0.314** | **0.217** | **0.282** |

Table XIII: **Performance comparison on the SumMe and TVSum datasets.** Results are reported under the 5-fold cross-validation protocol using the traditional train/validation (TV) setup.

Tab. XIII presents the performance comparison on the SumMe and TVSum datasets using the traditional train/validation (TV) split under the 5-fold cross-validation (5FCV) framework. These results are included primarily to ensure comparability with prior work, as this evaluation setting has been predominantly used in previous literature.

While the traditional TV split offers comparability with existing baselines and demonstrates the robustness of our approach across different random partitions, the TVT split provides a more reliable measure of true generalization performance by eliminating potential test set contamination during hyperparameter optimization. Across both evaluation settings, TripleSumm consistently surpasses previous methods, with particularly strong improvements when pre-trained on our MoSu dataset. This consistent performance gain across different evaluation protocols demonstrates the effectiveness of our multimodal fusion design and validates the benefits of large-scale pre-training data.

### F.2 Mr. HiSum Dataset

| Method | $\tau \uparrow$ | $\rho \uparrow$ | mAP50 $\uparrow$ | mAP15 $\uparrow$ |
|---|---|---|---|---|
| SUM-GAN (Mahasseni et al., 2017) | 0.067 | 0.095 | 56.62 | 23.56 |
| VASNet (Fajtl et al., 2018) | 0.069 | 0.102 | 58.69 | 25.28 |
| AC-SUM-GAN (Apostolidis et al., 2021a) | 0.012 | 0.018 | 55.35 | 21.88 |
| SL-module (Xu et al., 2021) | 0.060 | 0.088 | 58.63 | 24.95 |
| PGL-SUM (Apostolidis et al., 2021b) | 0.097 | 0.141 | 61.60 | 27.45 |
| Joint-VA (Badamdorj et al., 2021) | 0.161 | 0.231 | 65.88 | 35.23 |
| iPTNet (Jiang & Mu, 2022) | 0.020 | 0.029 | 55.53 | 22.74 |
| UMT (Liu et al., 2022) | 0.178 | 0.253 | 66.81 | 35.65 |
| A2Summ (He et al., 2023) | 0.121 | 0.172 | 63.20 | 32.34 |
| SSPVS (Li et al., 2023) | 0.078 | 0.113 | 59.48 | 26.35 |
| CSTA (Son et al., 2024) | 0.128 | 0.185 | 63.38 | 30.42 |
| **Ours** (Visual) | 0.187 | 0.258 | 67.16 | 35.57 |
| **Ours** (Full) | **0.258** | **0.352** | **70.72** | **40.88** |

Table XIV: **Comparison on Mr. HiSum dataset.**

To provide a complete benchmark, we report detailed results on the Mr. HiSum dataset in Tab. XIV. TripleSumm achieves the highest scores across all correlation metrics, surpassing previous state-of-the-art approaches by a clear margin.

## G DATASET ABLATION STUDY

### G.1 DATA SCALING

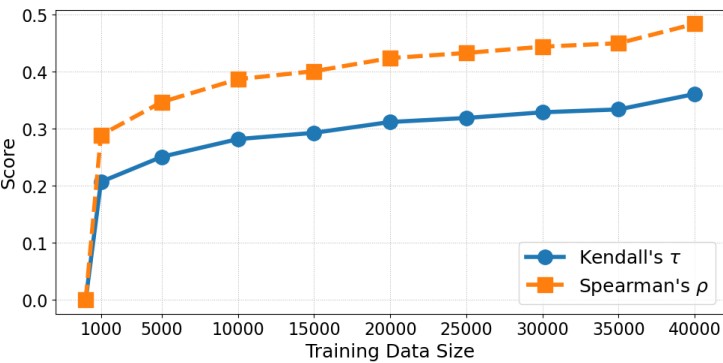

Figure III: **Evaluating the effect of MoSu dataset scale on TripleSumm performance.**

To examine the impact of training data volume on performance, TripleSumm was trained on MoSu subsets of progressively increasing size. As illustrated in Fig. III, both Kendall's $\tau$ and Spearman's $\rho$ exhibit a clear positive correlation with the number of training samples. Notably, the performance curve does not reach saturation, even when the entire MoSu dataset is utilized, which suggests that further performance gains could be achieved with additional data. This finding highlights the data scarcity of conventional benchmarks like SumMe and TVSum and underscores the necessity of large-scale datasets such as MoSu for advancing the field of video summarization.

### G.2 CLASS DISTRIBUTION

To assess label balance, the distribution of semantic clusters within the MoSu dataset was analyzed (App. B.3). The dataset is composed of ten distinct clusters of varying sizes and themes; however, the sample distribution is sufficiently balanced to prevent any single category from dominating the training process.

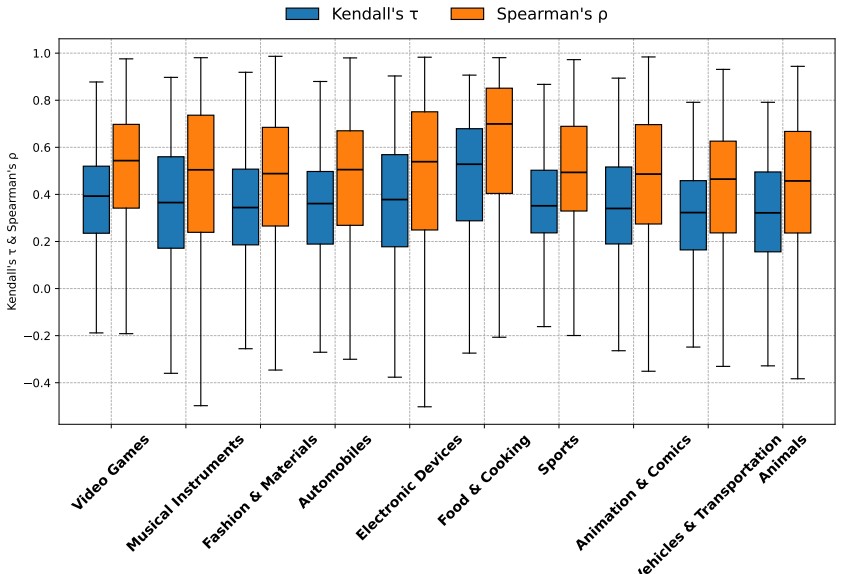

Figure IV: **Per-cluster performance on the MoSu dataset.** Boxplots show the distribution of Kendall's $\tau$ (blue) and Spearman's $\rho$ (orange) for each semantic cluster.

A per-cluster performance evaluation, measured by correlation metrics, further substantiates this balance (Fig. IV). The median scores across clusters remain proximal to the overall mean and exhibit no clear correlation with sample size. For instance, clusters with fewer samples often achieve performance comparable to that of larger ones, while some well-represented clusters show wider variance or slightly lower median scores. Notably, the 'Sports' and 'Animation & Comics' clusters achieve higher medians with tighter distributions, suggesting more consistent model predictions for these categories. These results suggest that model accuracy is more influenced by the difficulty of the content in each cluster (*e.g.*, visual complexity, audio variation) than by the amount of data.

## H    MORE QUALITATIVE RESULTS

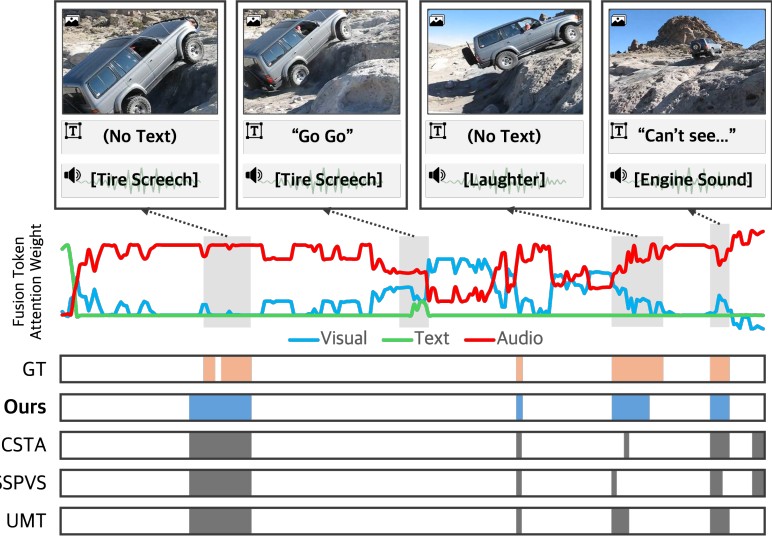

(a) A vehicle overcoming an obstacle, where the audio cue of a tire screech is critical to understanding the action.

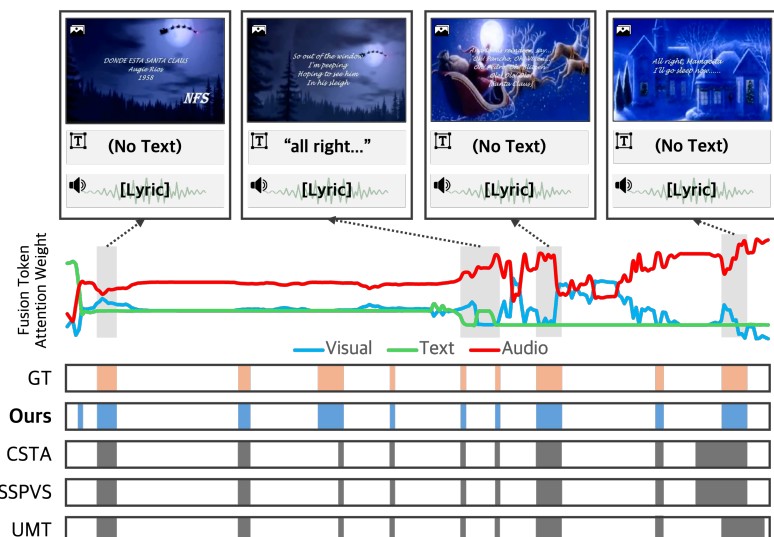

(b) A music video centered on its audio track, where visual and text cues are of secondary importance.

Figure V: **More Qualitative Examples on MoSu.** The graph in the middle visualizes the fusion token's attention weights, illustrating how our model dynamically estimates the saliency of each modality and thus maintains strong summarization accuracy even when some modalities are missing.

