# OpenReview forum: "TripleSumm: Adaptive Triple-Modality Fusion for Video Summarization"
_ICLR.cc/2026/Conference — ICLR 2026 Poster_

### Official Review · Reviewer_rpGf · 2025-10-17

**Soundness:** 3
**Presentation:** 3
**Contribution:** 2
**Rating:** 6
**Confidence:** 3

**Summary:**

TripleSumm introduces an adaptive tri-modal fusion architecture for video summarization that integrates visual, textual, and audio features at the frame level. The model interleaves two components: a Multi-Scale Temporal (MST) block, which captures temporal patterns with hierarchical windowed self-attention, and a Cross-Modal Fusion (CMF) block, which dynamically weighs modality importance using cross-attention. To support large-scale multimodal training, the authors also release MoSu, a new dataset of 52,678 videos featuring synchronized trimodal data and “Most Replayed” annotations from YouTube. TripleSumm achieves state-of-the-art results on four benchmarks (MoSu, Mr. HiSum, SumMe, and TVSum), outperforming prior unimodal and bimodal approaches by notable margins while remaining computationally efficient.

**Strengths:**

- The CMF block effectively learns frame-wise modality weighting, overcoming the common bias toward visual features in prior multimodal architectures.


- The MST block’s multi-scale attention enables both global context understanding and fine-grained temporal sensitivity.


- Extensive quantitative and qualitative evaluations across four datasets, plus thorough ablations (on modalities, window sizes, and fusion strategies), strengthen empirical claims.

**Weaknesses:**

- The adaptive fusion’s learning dynamics are empirically motivated but not theoretically analyzed (e.g., modality weighting stability, attention interpretability).


- Using “Most Replayed” statistics as pseudo-ground truth may encode platform or viewer biases, which are only partially mitigated.


- Despite its scale, MoSu is heavily drawn from YouTube-8M and English-transcribed videos, potentially limiting generalization to non-English or low-resource domains.


- The multi-scale temporal attention, though efficient per layer, could still pose latency issues for very long videos or real-time summarization.

**Questions:**

- How well would TripleSumm adapt to domains beyond YouTube-style videos, such as surveillance, documentaries, or multilingual news—where audio-text alignment and modality saliency differ significantly?

- How sensitive is performance to hyperparameters like window size, number of fusion layers, or modality embedding dimension? Would a learned or data-driven windowing schedule outperform the fixed [N,45,15,5] setup?

- Does the adaptive attention occasionally overfit to dominant modalities (e.g., always preferring audio in music videos)? Could regularization or entropy constraints improve stability?

- How do biases in the “Most Replayed” signal affect what the model learns to consider “important”? Has any qualitative audit been performed on viewer-driven biases (e.g., clickbait or thumbnail influence)?

- What is the real-time performance profile (e.g., inference time per video minute) compared to unimodal transformers, and could TripleSumm be deployed on edge devices or streaming platforms?

- Could the fusion and temporal mechanisms be generalized to other multimodal sequence tasks (e.g., event detection, captioning, or multimodal retrieval)? How would the model handle more than three modalities, such as motion or depth cues?

---

> ### Author Response · Authors · 2025-11-24
> **Response to Reviewer rpGf (1/5)**
>
> We appreciate the reviewers' comprehensive and insightful assessments. The raised questions provide an invaluable opportunity to significantly refine and clarify our paper. We answer each of them in detail below.
>
> ---
>
> __W1. Analysis of learning dynamics__
>
> __A.__ Following the reviewer’s suggestion, we provide additional analysis on the adaptive learning dynamics of our method in the revised Appendix E. We summarize our discovery below.
>
> First, we compare the performance degradation when we selectively take only a subset of modalities based on the per-frame modality saliency estimated by our model. We expect slight degradation when we drop only the least salient modality or take the strongest one only per-frame basis, while we expect substantial performance drop when we remove the modalities that are estimated as the most important ones. Table XII in Appendix E.1 verifies this: the summary accuracy gets significantly higher when the modalities with higher estimated saliency are included in the order. This implies that our model successfully captures the per-frame saliency of each modality.
>
> Second, we conduct a similar study as above, using a subset of modalities based on our estimated saliency, but with a fixed threshold instead of relative importance. As seen in Fig. II in Appendix E.2, we observe a smooth monotonically increasing curve when we drop modalities with their estimated saliency less than θ (Under θ), while another smooth monotonically decreasing curve when we drop modalities with their estimated saliency above a threshold θ (Over θ). These curves imply a stable learning dynamic where the model effectively utilizes nuanced, distributed saliency of each modality.
>
> ---
>
> __W2. Potential bias with the Most Replayed statistics__
>
> __A.__ We agree that the 'Most Replayed' (MR) statistic is a proxy for ground truth, not a perfectly unbiased annotation. We address this concern with the following three points.
>
> First of all, the use of an MR-like signal for summarization has been recently established in literature (e.g., Mr.HiSum [a], MRD [b]). This signal, while a proxy, is far less subjective and more generalizable than the small-scale human annotations in traditional datasets, thanks to its vast and collective nature of aggregating a significantly larger number of viewers (>50,000 viewers per video, in our case). In addition, we preprocess the signal to minimize temporal bias, as described in Appendix B.2.
>
> Also, our zero-shot and fine-tuned transfer experiments imply that our model trained on MoSu robustly performs well on human-annotated benchmarks like SumMe and TVSum in Table 3(b,c), or substantially longer and more complex videos in the revised Sec. 5.5. Such a transfer would not be possible if the Most-replayed-based labels are significantly biased.
>
> Lastly, we conduct a new human alignment study to directly measure the correlation between MR and human-annotated importance scores in the revised Appendix H.1. The rank correlations (Kendall’s $\tau$ and Spearman’s $\rho$) between the human-annotated scores and the MR scores are shown in the table below, exhibiting a positive correlation between them.
>
> | Metric                   | $\tau$     | $\rho$     |
> |--------------------------|-------|-------|
> | Human vs. MR Correlation | 0.466 | 0.602 |
>
> Putting them together, although we admit that MR stats may not be completely free from noise and bias, its benefit outweighs, serving as a suitable proxy of human judgement for video summarization.
>
> [a] Sul et al. Mr. HiSum: A Large-scale Dataset for Video Highlight Detection and Summarization. (NeurIPS 2024)
>
> [b] Kim et al. Generating Highlight Videos of a User-Specified Length using Most Replayed Data. (CHI 2025)

---

> ### Author Response · Authors · 2025-11-24
> **Response to Reviewer rpGf (2/5)**
>
> __W3. Non-English or low-resource domains in our dataset__
>
> __A.__ We would like to take this opportunity to clarify the scope of our data. First, our dataset is __not__ limited to English videos. As detailed in lines 859-887, non-English videos were translated to English using the official YouTube API. This decision was to include a diverse, multilingual, and global range of video content.
>
> Similarly, we were also mindful of potential topical bias. We chose YouTube 8M since its topical coverage is one of the widest among the public video sets, containing 3,406 classes. To maintain its thematic diversity as much as we can during the filtering process (e.g., excluding videos without MR stats or speech), we clustered the 3,406 fine-grained categories into 10 distinct, broad topical groups (e.g., Video Games, Musical Instruments, Food & Cooking, Sports), and performed stratified sampling, as detailed in Appendix B.3.
>
> While we acknowledge that this approach might still under-represent true low-resource domains (i.e., languages not supported by the API), our methodology represents a pragmatic and significant step toward capturing a more diverse global dataset.
>
> ---
>
> __W4. Latency for long videos__
>
> __A.__ We appreciate the reviewer for raising this important concern regarding computational efficiency. We agree that processing very long videos and real-time scenarios presents distinct challenges even with efficient windowed attention mechanisms.
>
> To demonstrate scalability and performance of our method on very long videos, we conducted an additional transfer experiment on 50 long videos (70.4 minutes on average, 10-20x longer than training videos), presented in the revised Sec 5.5. We compare the summary accuracy and inference latency of our method and the strongest unimodal baseline CSTA:
>
> | Computation Cost | $\tau$     | $\rho$     | GLOPs  | Latency |
> |------------|:------------:|:------------:|----------:|----------:|
> | CSTA             | 0.083 | 0.123 | 11.37G | 9.48ms  |
> | __TripleSumm__       | __0.128__ | __0.189__ | __0.97G__ | __2.81ms__ |
>
> As shown in the results, TripleSumm performs inference within a reasonable inference latency for real-time use, approximately 3-4 times faster than the baseline (CSTA). Together with the superior accuracy and latency measured on our main dataset in Appendix D, we conclude that our method is highly scalable without compromising accuracy.

---

> ### Author Response · Authors · 2025-11-24
> **Response to Reviewer rpGf (3/5)**
>
> __Q1. Generalizability to beyond YouTube-style videos__
>
> __A.__ Motivated by the reviewer's question, we conducted a new zero-shot experiment on 50 significantly longer (70.4 minutes-long on average) videos unseen at training, specifically designed to test TripleSumm's generalization to such complex domains. As detailed in the revised Sec. 5.5, our model achieves highly competitive performance, even if the test videos are substantially different from the training set in duration and complexity.
>
> ---
>
> __Q2. Sensitivity on hyperparameters__
>
> __A.__ Following the reviewer’s suggestion, we conducted additional ablation studies: window size (Appendix C.2), number of fusion layers (Appendix C.1), and modality embedding dimension (Appendix C.4). Our findings include that the model is somewhat sensitive to the temporal window sizing _strategy_, but far less sensitive to the particular size values within each strategy, as long as both local and global windows are sufficiently provided. Also, this additional study revealed that our method actually performs well with a much smaller embedding size than originally reported.
>
> We additionally apply adaptive (learned) windowing. Learning it from scratch is extremely unstable though, probably due to the excessive degree of freedom. Instead, we initialize the model with pre-trained ones using a fixed windowing schedule, and fine-tune them to see if it improves. The table below shows that this approach is indeed promising, slightly (but not significantly) improving the accuracy in most cases. We detail this in the revised Appendix C.3.
>
> | Method          | $\tau$              | $\rho$              | mAP50         | mAP15         |
> |-----------------|----------------|----------------|---------------|---------------|
> | Standard SA     | 0.327          | 0.442          | 73.84         | 42.74         |
> |     *+ Learnable* | 0.329 (+0.002) | 0.446 (+0.004) | 74.04 (+0.20) | 42.94 (+0.20) |
> | Global-to-Local | 0.354          | 0.474          | 74.75         | 44.71         |
> |     *+ Learnable* | 0.359 (+0.005) | 0.479 (+0.005) | 75.00 (+0.25) | 44.89 (+0.18) |
> | Local-to-Global | 0.361          | 0.484          | 75.11         | 45.53         |
> |     *+ Learnable* | 0.362 (+0.001) | 0.482 (-0.002) | 75.24 (+0.13) | 45.08 (-0.45) |

---

> ### Author Response · Authors · 2025-11-24
> **Response to Reviewer rpGf (4/5)**
>
> __Q3. Stability regarding overfitting to particular modalities__
>
> __A.__ This is an interesting and insightful question. Our model is indeed designed to prevent such an overfitting. Most existing methods use the visual [c] or text [d] features as the query to the fusion transformer, but we claim that such a design puts excessive emphasis on the particular modality and leads to overfitting on it. We replace this with an aggregated feature of all participating modalities, as described in lines 185-192. Thus, our model performs inference based on unbiased aggregate features, putting higher weights on truly relevant modalities per-example basis. This prevents the overfitting that the reviewer raised concern.
>
> We have experimental evidence verifying this. Qualitatively, examples in Fig. 3 and V illustrate how our model dynamically assigns modality importance based on the input, not biased to any prior. Quantitatively, we measure the summary accuracy by video themes in Appendix G.2. Our model performs equally well across all topics, including musical instruments (where audio modality would be mostly useful) and food & cooking (where text is particularly important).
>
>
> [c] Narasimhan et al. Clip-it! language-guided video summarization. (NeurIPS 2021)
>
> [d] Guo et al. CFSum: A Transformer-Based Multi-Modal Video Summarization Framework With Coarse-Fine Fusion. (ICASSP 2025)
>
> ---
>
> __Q4. Potential bias with the Most Replayed statistics__
>
> __A.__ We agree that the 'Most Replayed' (MR) statistic is a proxy for ground truth, not a perfectly unbiased annotation. We address this concern with the following three points.
>
> First of all, the use of an MR-like signal for summarization has been recently established in literature (e.g., Mr.HiSum [e], MRD [f]). This signal, while a proxy, is far less subjective and more generalizable than the small-scale human annotations in traditional datasets, thanks to its vast and collective nature of aggregating a significantly larger number of viewers (>50,000 viewers per video, in our case). In addition, we preprocess the signal to minimize temporal bias, as described in Appendix B.2. Also, we believe ‘Most Replayed’ is robust to clickbait or thumbnail influence as well, since it counts only when the part has been “replayed”, not simply “viewed”. If a viewer is attracted by clickbait but finds the content disappointing, they are unlikely to rewind that segment.
>
> Also, our zero-shot and fine-tuned transfer experiments imply that our model trained on MoSu robustly performs well on human-annotated benchmarks like SumMe and TVSum in Table 3(b,c), or substantially longer and more complex videos in the revised Sec. 5.5. Such a transfer would not be possible if the Most-replayed-based labels are significantly biased.
>
> Lastly, we conduct a new human alignment study to directly measure the correlation between MR and human-annotated importance scores in the revised Appendix H.1. The rank correlations (Kendall’s $\tau$ and Spearman’s $\rho$) between the human-annotated scores and the MR scores are shown in the table below, exhibiting a positive correlation between them.
>
> | Metric                   | $\tau$     | $\rho$     |
> |--------------------------|-------|-------|
> | Human vs. MR Correlation | 0.466 | 0.602 |
>
> Putting them together, although we admit that MR stats may not be completely free from noise and bias, its benefit outweighs, serving as a suitable proxy of human judgement for video summarization.
>
> [e] Sul et al. Mr. HiSum: A Large-scale Dataset for Video Highlight Detection and Summarization. (NeurIPS 2024)
>
> [f] Kim et al. Generating Highlight Videos of a User-Specified Length using Most Replayed Data. (CHI 2025)

---

> ### Author Response · Authors · 2025-11-24
> **Response to Reviewer rpGf (5/5)**
>
> __Q5. Real-time performance and streaming__
>
> __A.__ We provide a comprehensive computational cost analysis in Appendix D and summarized below, indicating that TripleSumm achieves superior summarization accuracy at competitive computational cost.
>
>
> First, we added a comprehensive computational cost analysis in Appendix D, comparing our TripleSumm against strong baselines. The result is copied here:
>
> | Method     | $\tau$       | $\rho$       | Params    | GFLOPs    | Inference Time |
> |------------|:------------:|:------------:|----------:|----------:|---------------:|
> | VASNet     | 0.151     | 0.219     | 8.31M     | 1.99G     | 7.36ms         |
> | PGL-SUM    | 0.151     | 0.218     | 5.31M     | 1.21G     | 13.55ms        |
> | CSTA       | 0.291     | 0.398     | 10.56M    | 11.37G    | 9.48ms         |
> | A2Summ     | 0.181     | 0.257     | 2.48M     | 1.35G     | 44.29ms        |
> | SSPVS      | 0.190     | 0.271     | 112.81M   | 43.64G    | 14.27ms        |
> | Joint-VA   | 0.190     | 0.272     | 4.21M     | 1.63G     | **2.58ms**     |
> | UMT        | 0.239     | 0.334     | 4.66M     | 1.39G     | 4.94ms         |
> | CFSum      | 0.277     | 0.374     | 19.83M    | 8.52G     | 3.87ms         |
> | **TripleSumm** | **0.361** | **0.484** | **1.37M** | **0.97G** | 2.81ms         |
>
> As shown in the table, TripleSumm uses only 1.37M learnable parameters, significantly more compact than all other baselines. Furthermore, despite processing four feature streams (Fusion, Visual, Text, Audio), our model requires the lowest computational operations (0.97 GFLOPs) among all competing methods. The inference time per video for our method is 2.81 ms, which is faster than most other methods, including UMT (4.94 ms), confirming its strong real-time performance capability.
>
> Based on this efficiency profile, TripleSumm is highly suitable for deployment on edge devices and streaming platforms. The small memory footprint afforded by the low parameter count makes it suitable for memory-constrained edge devices, and the combination of low latency (2.81 ms) and minimal computational cost (0.97 GFLOPs) makes it ideal for streaming platforms that require fast response times and efficient hardware utilization.
>
> ---
>
> __Q6. Extension to other multimodal sequence tasks and additional modalities__
>
> __A.__ Although our current prediction head is specialized for summarization, the core MST and CMF modules can be potentially adapted to other tasks that require frame-level scoring, such as event detection or highlight localization. Moreover, the refined fused embeddings can serve as an input to a text decoder for captioning or as embeddings for multimodal retrieval.
>
> To assess extensibility beyond three modalities, we add I3D motion features as the fourth one and train our model on a subset of the MoSu with 5,000 videos. (We had to do this with a subset due to the high feature extraction cost during this short rebuttal period. We will update this in the camera-ready.) In the table below, we observe a significant boost of summarization accuracy, demonstrating that additional complementary signals are naturally absorbed by the fusion mechanism.
>
> | Model      | Params | $\tau$     | $\rho$     | mAP50 | mAP15 |
> |------------|--------|-------|-------|-------|-------|
> | Ours       | 1.37M  | 0.213 | 0.300 | 67.09 | 33.08 |
> | Ours + I3D | 1.51M  | 0.253 | 0.349 | 68.69 | 36.45 |

---

### Official Review · Reviewer_dGFC · 2025-10-19

**Soundness:** 2
**Presentation:** 2
**Contribution:** 2
**Rating:** 4
**Confidence:** 3

**Summary:**

TripleSumm proposes an adaptive multimodal video summarization framework that integrates visual, audio, and textual cues through a Multi-Scale Temporal Block and a Cross-Modal Fusion Block. It also builds a new large-scale trimodal video summarization dataset with aligned video, transcript, and audio data. Experiments show that the method effectively models both temporal and multimodal patterns, achieving strong summarization performance.

**Strengths:**

- S1: The separation between temporal modeling (MST) and cross-modal fusion (CMF) is elegant and easy to interpret.
- S2: The framework effectively captures both intra-modal temporal dependencies and inter-modal relationships.
- S3: The new dataset provides valuable resources for future research on multimodal summarization.

**Weaknesses:**

- W1: This paper shows relatively weak originality. The proposed model mainly focuses on optimizing multimodal (visual, text, audio) feature representations through standard attention operations, where self-attention is used to enhance intra-modal features and cross-attention is used for inter-modal fusion. The motivation and methodology closely resemble those of earlier works such as UMT [1] and CFSum [2], without demonstrating a clear conceptual or technical advancement beyond them.

- W2: This paper’s methodological contributions lie almost entirely in representation learning, without offering task-specific innovations for video summarization itself. The proposed feature enhancement techniques could be applied broadly to other video-related tasks such as action recognition, rather than being tailored to the summarization objective. Moreover, this work is not the first to integrate three modalities (visual, textual, and audio), as prior studies have already explored trimodal fusion for similar purposes.

- W3: The “multi-scale” temporal modeling relies on manually predefined window sizes (e.g., N, 45, 15, 5) rather than adaptive or learnable scales. This heuristic choice may not generalize well to videos of different lengths or dynamics, limiting flexibility and scalability.

- W4: The paper does not include detailed ablation studies isolating the effects of each module (e.g., MST vs. CMF, scale depth, or fusion token design).

[1] UMT: Unified Multi-modal Transformers for Joint Video Moment Retrieval and Highlight Detection

[2] CFSum: A Transformer-Based Multi-Modal Video Summarization Framework With Coarse-Fine Fusion

**Questions:**

- Q1: Is the reported performance overly dependent on the newly introduced MoSu dataset, whose annotation process (based on replay statistics) may introduce bias and limit reproducibility, potentially misaligning with true human summarization preferences?

- Q2: Is there any discussion of computational cost, such as runtime or FLOPs comparisons, to support the claimed efficiency of the method?

---

> ### Author Response · Authors · 2025-11-24
> **Response to Reviewer dGFC (1/3)**
>
> We sincerely thank the reviewer for meticulous and insightful critique. We are grateful for the opportunity to address key concerns regarding the conceptual novelty and task-specific design of TripleSumm, as well as the validation methodology for our architecture and the MoSu dataset. The questions and constructive feedback are indeed valuable for us to improve our paper, as detailed below.
>
> ---
>
> __W1. Originality__
>
> __A.__ As the reviewer commented, we do not claim novelty on the self/cross-attention operations themselves, which have been established in many other works. The true originality of our work lies in architecturally achieving a fundamental, unaddressed problem specific to video summarization: _dynamically adapting to salient modalities_. Unlike UMT and CFSum, which employ static or simple fusion for general multimodal representation, video summarization requires capturing the frame-level temporal shift in which modality is most salient (e.g., visual vs. text vs. audio). To the best of our knowledge, our Adaptive Fusion Mechanism is the first architecture explicitly designed to model and dynamically adapt to this rapidly shifting saliency at every frame, enabling superior performance by focusing on the most relevant signal at any given moment. We illustrate this in Fig. 3 and Appendix I, and revise the Introduction (lines 45-53) to more clearly reflect this motivation. Please see our answer to W2 below as well.
>
> ---
>
> __W2. Methodological contribution specific to video summarization__
>
> __A.__ We respectfully disagree with the reviewer’s comment that our methodological design is not specific to video summarization. A fundamental challenge in multimodal video summarization is the dynamic shift of frame-level importance of each modality, as we have motivated in Fig. 1. Although we acknowledge prior multimodal fusion methods, no existing video summarization architecture has dynamically adapted to moment-to-moment saliency per modality at frame level; rather, they have relied on static fusion, leveraging only the mutually compensating information in different modalities. On the other hand, our separated but interleaved Multi-scale Temporal block and Cross-modal Fusion block effectively help the model to capture the most dominant modality at any given time. We claim that this design is particularly useful for summarization, where frames and modalities tend to _compete_ for higher importance, by properly controlling where to attend among multimodal features and nearby frames alternatively. This is in contrast to other tasks like video classification where multimodal features tend to _cooperate_ to predict the correct label, where a simple fusion or global attention can be still beneficial whenever computational resources and data are sufficiently provided. The superior performance achieved by our method indicates its effectiveness in this particular task, where no other general video understanding has been simply applied to tackle. We reflected this argument in the revised lines 45-53 and 132-139.

---

> ### Author Response · Authors · 2025-11-24
> **Response to Reviewer dGFC (2/3)**
>
> __W3. Manually selected window sizes__
>
> __A.__ Following the reviewer’s great suggestion, we additionally apply adaptive (learned) windowing. Learning it from scratch is extremely unstable though, probably due to the excessive degree of freedom. Instead, we initialize the model with pre-trained ones using a fixed windowing schedule, and fine-tune them to see if it improves. The table below shows that this approach is indeed promising, slightly (but not significantly) improving the accuracy in most cases. We detail this in the revised Appendix C.3.
>
> | Method          | $\tau$              | $\rho$              | mAP50         | mAP15         |
> |-----------------|----------------|----------------|---------------|---------------|
> | Standard SA     | 0.327          | 0.442          | 73.84         | 42.74         |
> |     *+ Learnable* | 0.329 (+0.002) | 0.446 (+0.004) | 74.04 (+0.20) | 42.94 (+0.20) |
> | Global-to-Local | 0.354          | 0.474          | 74.75         | 44.71         |
> |     *+ Learnable* | 0.359 (+0.005) | 0.479 (+0.005) | 75.00 (+0.25) | 44.89 (+0.18) |
> | Local-to-Global | 0.361          | 0.484          | 75.11         | 45.53         |
> |     *+ Learnable* | 0.362 (+0.001) | 0.482 (-0.002) | 75.24 (+0.13) | 45.08 (-0.45) |
>
> ---
>
> __W4. Ablation on each module__
>
> __A.__ We conducted the suggested ablation study to verify the stand-alone effectiveness of CMF and MST blocks. The revised Table 4(c) (copied below) confirms that the two blocks play their own role for the improved performance.
>
> | MST | CMF | $\tau$     | $\rho$     | mAP50 | mAP15 |
> |:---:|:---:|:----------:|:----------:|:-----:|:-----:|
> |     | ✓   | 0.260 | 0.363 | 70.40 | 38.46 |
> | ✓   |     | 0.346 | 0.464 | 74.28 | 44.21 |
> | ✓   | ✓   | __0.361__ | __0.484__ | __75.11__ | __45.53__ |
>
> The other two suggested ablation studies were provided in the original manuscript: scale depth in Appendix C.1 and fusion token design in the revised Appendix C.6 (originally was in Table 4(d)).

---

> ### Author Response · Authors · 2025-11-24
> **Response to Reviewer dGFC (3/3)**
>
> __Q1. Potential bias with the Most Replayed statistics__
>
> __A.__ We agree that the 'Most Replayed' (MR) statistic is a proxy for ground truth, not a perfectly unbiased annotation. We address this concern with the following three points.
>
> First of all, the use of an MR-like signal for summarization has been recently established in literature (e.g., Mr.HiSum [a], MRD [b]). This signal, while a proxy, is far less subjective and more generalizable than the small-scale human annotations in traditional datasets, thanks to its vast and collective nature of aggregating a significantly larger number of viewers (>50,000 viewers per video, in our case). In addition, we preprocess the signal to minimize temporal bias, as described in Appendix B.2.
>
> Also, our zero-shot and fine-tuned transfer experiments imply that our model trained on MoSu robustly performs well on human-annotated benchmarks like SumMe and TVSum in Table 3(b,c), or substantially longer and more complex videos in the revised Sec. 5.5. Such a transfer would not be possible if the Most-replayed-based labels are significantly biased.
>
> Lastly, we conduct a new human alignment study to directly measure the correlation between MR and human-annotated importance scores in the revised Appendix H.1. The rank correlations (Kendall’s $\tau$ and Spearman’s $\rho$) between the human-annotated scores and the MR scores are shown in the table below, exhibiting a positive correlation between them.
>
> | Metric                   | $\tau$     | $\rho$     |
> |--------------------------|-------|-------|
> | Human vs. MR Correlation | 0.466 | 0.602 |
>
> Putting them together, although we admit that MR stats may not be completely free from noise and bias, its benefit outweighs, serving as a suitable proxy of human judgement for video summarization.
>
> [a] Sul et al. Mr. HiSum: A Large-scale Dataset for Video Highlight Detection and Summarization. (NeurIPS 2024)
>
> [b] Kim et al. Generating Highlight Videos of a User-Specified Length using Most Replayed Data. (CHI 2025)
>
> ---
>
> __Q2. Computational cost__
>
> __A.__ Following the reviewer’s suggestion, we added a comprehensive computational cost analysis in Appendix D, comparing our TripleSumm against strong baselines. The result is copied here:
>
> | Method     | $\tau$       | $\rho$       | Params    | GFLOPs    | Inference Time |
> |------------|:------------:|:------------:|----------:|----------:|---------------:|
> | VASNet     | 0.151     | 0.219     | 8.31M     | 1.99G     | 7.36ms         |
> | PGL-SUM    | 0.151     | 0.218     | 5.31M     | 1.21G     | 13.55ms        |
> | CSTA       | 0.291     | 0.398     | 10.56M    | 11.37G    | 9.48ms         |
> | A2Summ     | 0.181     | 0.257     | 2.48M     | 1.35G     | 44.29ms        |
> | SSPVS      | 0.190     | 0.271     | 112.81M   | 43.64G    | 14.27ms        |
> | Joint-VA   | 0.190     | 0.272     | 4.21M     | 1.63G     | **2.58ms**     |
> | UMT        | 0.239     | 0.334     | 4.66M     | 1.39G     | 4.94ms         |
> | CFSum      | 0.277     | 0.374     | 19.83M    | 8.52G     | 3.87ms         |
> | **TripleSumm** | **0.361** | **0.484** | **1.37M** | **0.97G** | 2.81ms         |
>
> First, TripleSumm uses only 1.37M _learnable parameters_, significantly more compact than all other baselines. This high efficiency is largely thanks to the parameter sharing strategy in our Multi-scale Temporal (MST) block.
>
> Despite processing four feature streams (Fusion, Visual, Text, Audio) through multiple blocks, our TripleSumm maintains remarkably low _computational complexity_, requiring the least amount of operations (0.97GFLOPs) for single video inference among the competing methods. The _inference time per video_ for our method (2.81 ms) is also faster than most other methods, following Joint-VA. This analysis concludes that our TripleSumm achieves superior accuracy ($\tau=0.361, \rho=0.484$) with an extremely competitive computational cost, addressing the reviewer's concern regarding latency and efficiency.

---

### Official Review · Reviewer_hyzT · 2025-10-30

**Soundness:** 3
**Presentation:** 3
**Contribution:** 3
**Rating:** 6
**Confidence:** 5

**Summary:**

This paper tackles multimodal video summarization by proposing TripleSumm, an architecture that performs frame-level adaptive fusion of three modalities. To address the widely acknowledged shortage of suitable evaluation resources in this area, the authors also introduce MoSu, a new large-scale benchmark providing synchronized tri-modal data. Experimental results show that TripleSumm yields clear state-of-the-art gains across four benchmarks, including MoSu, indicating the effectiveness of the fusion strategy and the value of the new dataset.

**Strengths:**

1. Well-motivated architecture design — TripleSumm performs adaptive, frame-level fusion of visual, audio, and text signals using specialized Modality and Temporal Blocks, enabling both fine-grained and long-range semantic capture.
2. Benchmark contribution — MoSu fills a critical gap by offering the first large-scale trimodal dataset for video summarization, which meaningfully advances evaluation and reproducibility in this field.
3. Strong empirical results — TripleSumm achieves state-of-the-art performance on four public benchmarks (including MoSu) with notable improvements and does so without excessive model size, demonstrating both effectiveness and efficiency.

**Weaknesses:**

1. Dataset Details and Quality Control
 The paper should include more comprehensive details about the proposed dataset, such as the average and variance of summary lengths, textual and audio statistics, and the distribution of video durations. Moreover, the authors need to justify why the generated summaries can be considered high-quality representations of the original videos under the current construction pipeline. It is strongly recommended that the summary quality be validated through human evaluation to ensure reliability.
2. Model Design Rationale
 Some architectural design choices lack clear justification. For instance, the model employs larger temporal window sizes in the early layers and smaller ones in the later layers. This design appears counterintuitive, as shallow layers typically lack the capacity to model long-range temporal dependencies. The authors should clarify the motivation behind this choice or consider the more conventional approach of using smaller windows in early layers and progressively larger ones in deeper layers.
3. Clarity of Figure 2
 Figure 2 is somewhat misleading. The main text states that the four modalities are processed independently by the MST, whereas the figure depicts them as being input jointly, suggesting possible cross-modal interaction within MST. The authors should revise the figure or clarify in the text how the modalities are actually handled to avoid confusion.
4. Generalization Evaluation
 To better demonstrate the generalization capability of the model and the utility of the proposed dataset, it is recommended to train the model on the new dataset and evaluate it on other benchmark datasets (with or without fine-tuning). Such experiments would significantly enhance the credibility and impact of the proposed dataset within the research community.

**Questions:**

Please refer to the Weaknesses section.

---

> ### Author Response · Authors · 2025-11-24
> **Response to Reviewer hyzT (1/1)**
>
> We appreciate the reviewer's detailed feedback. We provide comprehensive dataset statistics with human validation, clarify the rationale behind our Multi-scale Temporal block design, provide clarification regarding Figure 2, and include extensive generalization and transfer learning experiments.
>
> ---
>
> __W1. Dataset details and quality control__
>
> __A.__ Following the reviewer's suggestion, we provide more detailed stats of MoSu in Table II in the revised Appendix B.1, supplementing the overview in Sec. 4. As requested, this revision now includes specific breakdown of video durations, textual and audio statistics. Additionally, the summary length for each video is set to 15% of the original video's duration.
>
> | Category | Statistic             | Value               |
> |----------|-----------------------|---------------------|
> | Duration | Avg. Duration         | 272.25 sec          |
> |          | Std. Deviation        | 102.43 sec          |
> |          | Min / Max             | 120.00 / 501.00 sec |
> | Textual  | Total # of Tokens     | 32.6M               |
> |          | Avg. Tokens per Video | 619.1               |
> |          | Transcript Density    | 61.84%              |
> | Audio    | Audio Availability    | 100% (Filtered)     |
>
> Regarding the reliability of the ‘Most Replayed (MR)’ stats, we fully acknowledge the reviewer's concern and conducted human evaluation to directly verify the correlation between MR and human-annotated importance scores on a subset of the MoSu test set (see the revised Sec. H.1 for details). The rank correlations (Kendall’s $\tau$ and Spearman’s $\rho$) between the human-annotated scores and the MR scores are shown in the table below, exhibiting a positive correlation between them. This suggests that although Most Replayed is only a proxy, it maintains a positive alignment with human judgments of importance, supporting its viability as a supervision signal for video summarization.
>
> | Metric                   | $\tau$     | $\rho$     |
> |--------------------------|-------|-------|
> | Human vs. MR Correlation | 0.466 | 0.602 |
>
> ---
>
> __W2. Model design rationale: smaller-to-larger window size__
>
> __A.__ We sincerely thank the reviewer for this insightful feedback. In our original submission, we adopted a global-to-local design, placing larger temporal windows in early layers and smaller ones in later layers, intending to capture broad temporal cues as early context. However, as the reviewer correctly pointed out, lower layers may not possess sufficient representational capacity to effectively leverage such long-range temporal information. Following the reviewer’s suggestion, we experiment with the opposite (local-to-global) design, and achieve even superior performance. We report this experimental result in the revised lines 461-475 and Table 4(b), indicating that the new Local-to-Global even outperforms our previous best, Global-to-Local. The reviewer’s comment indeed provided a valuable insight to complete our model design, to fully leverage its potential. We added a brief discussion about this design rationale in lines 222-224.
>
> ---
>
> __W3. Clarification on Fig. 2__
>
> A. We thank the reviewer for pointing this out. We originally intended to emphasize the use of shared parameters across the Modality-Specific Transformer (MST) blocks, but agree that this would have been confusing regarding cross-modal interaction. We revised Fig. 2 as suggested to clearly demonstrate the independent processing of each modality.
>
> ---
>
> __W4. Generalization evaluation__
>
> __A.__ We thank the reviewer again for this insightful suggestion. We first remind the reviewer that we already had the suggested dataset transfer experiments in Table 3(b,c), on SumMe and TVSum, respectively. Our model pre-trained on MoSu achieved state-of-the-art performance on these sets when transferred. To further demonstrate its generalizability, we conduct zero-shot evaluation of our model trained on MoSu on 50 significantly (10-20x) longer videos unseen at training. As detailed in the new Sec. 5.5, our model achieves highly competitive performance, even if the test videos are substantially different from the training set in duration and complexity. From these two experiments, we conclude that our model is robustly generalizable to different datasets.

---

### Official Review · Reviewer_bwyx · 2025-10-31

**Soundness:** 3
**Presentation:** 3
**Contribution:** 3
**Rating:** 6
**Confidence:** 4

**Summary:**

This paper proposes TripleSumm, a trimodal video summarization framework that adaptively fuses visual, textual, and audio information to better capture the dynamics of real-world videos. The authors argue that existing methods are limited by unimodal or bimodal inputs and cannot adapt when the dominant modality changes over time. TripleSumm introduces two key components: a Multi-scale Temporal (MST) block that progressively refines attention from global to local temporal contexts, and a Cross-modal Fusion (CMF) block that computes per-frame adaptive weights to determine the relative importance of each modality. To support large-scale multimodal learning, the authors also construct MoSu (Most Replayed Multimodal Video Summarization), the first large trimodal dataset containing over 52,000 YouTube videos with synchronized visual, text, and audio signals and “most replayed” statistics as pseudo-ground-truth for frame importance. Extensive experiments across four benchmarks (MoSu, Mr. HiSum, SumMe, and TVSum) demonstrate that TripleSumm achieves state-of-the-art results with strong generalization and efficiency, supported by comprehensive ablations validating its dynamic fusion design and multi-scale temporal modeling.

**Strengths:**

Clear, modular architecture that explicitly separates temporal refinement (MST) from cross-modal fusion (CMF).

 Strong, consistent gains across multiple datasets/metrics with a small model size.

 Careful ablation validating design choices (windowing schedule, dynamic fusion, modality combos).

**Weaknesses:**

Reliance on “Most Replayed” as ground truth, while pragmatic, can encode popularity/behavior biases; human alignment on MoSu isn’t quantified beyond transfers to SumMe/TVSum. It would be better to human-study agreement or correlation with editorial summaries.

 Selection uses standard KTS + knapsack; could the model be trained end-to-end with a differentiable or learning-to-select objective, and would that change results?

 While parameter-efficient, the inference cost with four MST blocks and CMF per frame isn’t fully benchmarked vs. strong baselines at equal latency. A runtime/memory comparison would help.

**Questions:**

For MoSu, non-English transcripts are machine-translated; for other sets, text is generated via image captioning. How sensitive are results to transcript quality and captioning choice? Any robustness analysis?

---

> ### Author Response · Authors · 2025-11-24
> **Response to Reviewer bwyx (1/2)**
>
> We sincerely thank the reviewers for their insightful and constructive comments. We address the questions regarding the ground truth reliability, selection methodology, computational efficiency, and modality robustness below.
>
> ---
>
> __W1. Human alignment of “Most Replayed” as GT__
>
> __A.__ We fully acknowledge the reviewer's concern regarding the potential biases inherent in using the "Most Replayed" (MR) metric. Following the reviewer’s suggestion, we conducted a user study to directly measure the correlation between MR and human-annotated importance scores on a subset of the MoSu test set (see the revised Appendix H.1 for details).
>
> The rank correlations (Kendall’s $\tau$ and Spearman’s $\rho$) between the human-annotated scores and the MR scores are shown in the table below, exhibiting a positive correlation between them. This confirms that while Most Replayed is a proxy, it is still positively aligned with human judgements of importance, validating its use as a supervision source for video summarization.
>
> | Metric                   | $\tau$     | $\rho$     |
> |--------------------------|-------|-------|
> | Human vs. MR Correlation | 0.466 | 0.602 |
>
> ---
>
> __W2. Possibility of end-to-end training instead of KTS + knapsack__
>
> __A.__ The reviewer raises an excellent point regarding the potential benefits of an end-to-end, learning-to-select objective. While end-to-end selection such as differentiable selection sounds conceptually appealing, their applicability has been limited primarily by the lack of labeled data for video summarization. Due to the inherent subjectivity of the task, there is no single true summary and most datasets provide only frame-level importance scores rather than complete ground-truth summaries. As a result, the community focused on the reproducible part, predicting these frame scores, while leaving the selection to deterministic algorithms like KTS and knapsack. To ensure fair comparison with prior work and to isolate the contribution of our tri-modal scoring architecture, we follow this established procedure. An end-to-end selection remains as an interesting and valuable direction for our future work, and we revised the Conclusion section to reflect this.
>
> ---
>
> __W3. Runtime/memory comparison__
>
> __A.__ Following the reviewer’s suggestion, we added a comprehensive computational cost analysis in Appendix D, comparing our TripleSumm against strong baselines. The result is copied here:
>
> | Method     | $\tau$       | $\rho$       | Params    | GFLOPs    | Inference Time |
> |------------|:------------:|:------------:|----------:|----------:|---------------:|
> | VASNet     | 0.151     | 0.219     | 8.31M     | 1.99G     | 7.36ms         |
> | PGL-SUM    | 0.151     | 0.218     | 5.31M     | 1.21G     | 13.55ms        |
> | CSTA       | 0.291     | 0.398     | 10.56M    | 11.37G    | 9.48ms         |
> | A2Summ     | 0.181     | 0.257     | 2.48M     | 1.35G     | 44.29ms        |
> | SSPVS      | 0.190     | 0.271     | 112.81M   | 43.64G    | 14.27ms        |
> | Joint-VA   | 0.190     | 0.272     | 4.21M     | 1.63G     | **2.58ms**     |
> | UMT        | 0.239     | 0.334     | 4.66M     | 1.39G     | 4.94ms         |
> | CFSum      | 0.277     | 0.374     | 19.83M    | 8.52G     | 3.87ms         |
> | **TripleSumm** | **0.361** | **0.484** | **1.37M** | **0.97G** | 2.81ms         |
>
> First, TripleSumm uses only 1.37M _learnable parameters_, significantly more compact than all other baselines. This high efficiency is largely thanks to the parameter sharing strategy in our Multi-scale Temporal (MST) block.
>
> Despite processing four feature streams (Fusion, Visual, Text, Audio) through multiple blocks, our TripleSumm maintains remarkably low _computational complexity_, requiring the least amount of operations (0.97GFLOPs) for single video inference among the competing methods. The _inference time per video_ for our method (2.81 ms) is also faster than most other methods, following Joint-VA. This analysis concludes that our TripleSumm achieves superior accuracy ($\tau=0.361, \rho=0.484$) with an extremely competitive computational cost, addressing the reviewer's concern regarding latency and efficiency.

---

> ### Author Response · Authors · 2025-11-24
> **Response to Reviewer bwyx (2/2)**
>
> __Q1. Sensitivity on transcript quality and captioning choice__
>
> __A.__ We appreciate the reviewer for this insightful question regarding the impact of text modality. We demonstrate the robustness of our method to text quality and sources through our cross-domain transfer experiment. As shown in Table 2 and 3, the model pretrained on MoSu (using _speech-based transcripts_) achieves state-of-the-art performance even when fine-tuned on SumMe and TVSum, which rely on generated _image captions_. This successful transfer across different text types strongly implies robustness of our method on captioning choice, effectively generalizing the semantic knowledge learned from noisy transcripts to visual-based captions.
>
> In addition, we present another interesting observation comparing the performance of our model, one with caption-based and another with transcript-based texts on 2118 videos from MoSu. As presented in the table below, transcript-based outperforms the caption-based, in spite of the noise in the transcript. We interpret this by the amount of information complementary to the other modalities; that is, the transcript (speech) often provides new information unavailable in the visual or audio sources, while the image captions always convey overlapping information with the visual features. This result indirectly verifies again that our method effectively captures useful information from the noisy transcripts whenever useful, achieving superior performance than when only redundant information is provided (caption).
>
> | Text Source        | $\tau$     | $\rho$     | mAP50 | mAP15 |
> |----------------------|-------|-------|-------|-------|
> | Ours w/ Caption      | 0.200 | 0.283 | 66.23 | 31.85 |
> | Ours w/ Transcript   | 0.237 | 0.330 | 67.95 | 33.71 |

---

### Author Response · Authors · 2025-12-01
**Rebuttal Summary for AC**

We sincerely appreciate AC for handling our submission under this unusual circumstance. Below, we summarize our answers to the major concerns and questions raised by the reviewers. Please see the corresponding rebuttals for more details.

---

__1. “Most Replayed” stats need to be further verified by human evaluation to be used as ground-truth. (bwyx, hyzT, dGFC, rpGf)__

- We conducted a user study (Appendix H.1) confirming a positive rank correlation between Most Replayed stats and human-annotated importance scores.

__2. Multiple reviewers (hyzT, dGFC, rpGf) suggested several ideas to further improve the model.__

- We adopt the “Local-to-Global” window schedule (hyzT), achieving further improved performance (Table 4b). We also try learnable window schemes (hyzT, dGFC, rgGf), reporting marginally better performance in Appendix C.3.

__3. Multiple reviewers (bwyx, dGFC, rpGf) raised questions regarding the computational efficiency of the proposed method.__

- We added a complexity analysis (Appendix D). Our TripleSumm is highly efficient (87% reduction of learnable parameters, 3.37x faster inference compared to the previous best model, CSTA) according to this study.

__4. Evaluation on a new dataset (hyzT) with significantly longer videos (rpGf) would be useful to prove the proposed model’s generalizability.__

- We conducted a zero-shot evaluation on 50 long-form videos (avg. 70.4 mins) in Sec. 5.5, demonstrating that our model consistently outperforms baselines on unseen domains.

__5. Multiple reviewers (bwyx, rpGf) questioned the sensitivity or stability of our model on various factors.__

- We empirically verify robustness on transcript quality and captioning choice (bwyx), on hyperparameters like window size, layers, and latent dimensions (rpGf; Appendix C), and against overfitting to a single modality (rpGf; Appendix G.2). We also show that our method is effectively extendible to additional modalities, e.g., motion (rpGf).

__6. Misc.__

- Regarding the critiques on originality specific to video summarization (dGFC), our model is the first adaptive modality fusion method for video summarization to the best of our knowledge, dynamically shifting modality saliency at the frame level. We added ablation study on MST or CMF blocks in Table 4c (dGFC).

---

### Meta-Review · Area_Chair_8952 · 2026-01-05

**Summary:**

This work proposes TripleSumm, a trimodal video summarization framework that adaptively fuses visual, textual, and audio information to better capture the dynamics of real-world videos. A Multi-scale Temporal (MST) block and a Cross-modal Fusion (CMF) block are proposed to enhance the temporal contexts and learning representation from three modalities. Besides, a new benchmark MoSu is proposed with three modalities in video summarization.

**Reviewer Concerns:**

1. Dataset Details and Quality Control are addressed.
2. Ablation studies are addressed.
3. Misleading in some tables and figures is addressed.
4.  Technique contributions are relatively addressed but still outstanding.
Overall, most of concerns from reveiwers are addressed in rebuttal.

**Reviewer Scores:**

The reviewer dGFC give nagetive comments and the other three reviewers give positive score.The main concern from reviewer dGFC is about technique contributions and originality, and the author give relatively clear responses to the reviewer dGFC.

---

### Decision · Program_Chairs · 2026-01-26

Accept (Poster)